# Computed Tomographic and Histopathologic Studies of Lung Function Immediately *Post Natum* in Canine Neonates

**DOI:** 10.3390/ani13111741

**Published:** 2023-05-24

**Authors:** Jens Peter Teifke, Cornelia Peuckert, Jens-Christian Rudnick, Kathrin Büttner, Hartwig Bostedt

**Affiliations:** 1Friedrich-Loeffler-Institut, Federal Research Institute for Animal Health, 17493 Greifswald-Insel Riems, Germany; 2Veterinary Clinic Rostock GmbH, 18059 Rostock, Germany; 3Clinic for Obstetrics, Gynecology and Andrology of Large and Small Animals with Veterinary Ambulance, Justus-Liebig-University, 35392 Giessen, Germany; 4AG Biomathematics and Data Processing, Department of Veterinary Medicine, Justus-Liebig-University, 35392 Giessen, Germany

**Keywords:** canine neonates, computer tomography, histopathology, lung morphometry

## Abstract

**Simple Summary:**

The aim of this study was to analyze the spatial and temporal progress of the aeration of lung tissue in vital canine neonates during the first day post natum (p.n.). To this end, computed tomography (CT) was used, and histopathology was applied to quantify the area of aerated neonatal lung tissue. It was shown that, within the first 10 min p.n., the degree of ventilation in vital born pups reached about 75% of the final values obtained 24 h p.n. The dorsal lung areas were always significantly better ventilated than the ventral regions. The results of this study are clinically relevant and suggest that resuscitation measures should be performed already within the first 10 min p.n., preferably with the pup in the chest–abdomen position, to achieve the best ventilation of the lungs.

**Abstract:**

Background: The lung tissue in newborn canine neonates is still in a morphologically and functionally immature, canalicular–saccular stage. True alveoli are only formed postnatally. The aim of this study was to analyze the spatial and temporal development of the ventilation of the lung tissue in vital canine neonates during the first 24 h post natum (p.n.). Methods: Forty pups (birth weight Ø 424 g ± 80.1 g) from three litters of large dog breeds (>20 kg live weight) were included in the studies. Thirty-three pups (29 vital, 2 vitally depressed, 2 stillborn neonates) originated from controlled, uncomplicated births (*n* = 3); moreover, six stillborn pups as well as one prematurely deceased pup were birthed by other dams with delivery complications. Computed tomography (CT) was used in 39 neonates, and histopathologic tissue classification techniques (HALO) were used in 11 neonates (eight stillborn and three neonates died early post natum, respectively) to quantify the degree of aerated neonatal lung tissue. Results: It was shown that, in vital born pups, within the first 10 min p.n., the degree of ventilation reached mean values of −530 (±114) Hounsfield units (HU) in the dorsal and −453.3 (±133) HU in the ventral lung area. This is about 75–80% of the final values obtained after 24 h p.n. for dorsal −648.0 (±89.9) HU and ventral quadrants −624.7 (±76.8) HU. The dorsal lung areas were always significantly better ventilated than the ventral regions (*p* = 0.0013). CT as well as histopathology are suitable to clearly distinguish the nonventilated lungs of stillborns from neonates that were initially alive after surviving neonatal respiratory distress syndrome but who died prematurely (*p* = 0.0398). Conclusion: The results of this study are clinically relevant since the lung tissue of canine neonates presents an aeration profile as early as 10 min after birth and continues progressively, with a special regard to the dorsal lung areas. This is the basis for resuscitation measures that should be performed, preferably with the pup in the abdomen–chest position.

## 1. Introduction

The survival of a normally developed, homoiothermic individual in the first postnatal adaptation period depends primarily on whether the cardiopulmonary functional unit fully resumes its physiological activity immediately after expulsion from the uterine–vaginal space. If this is not the case, life-threatening conditions can occur, which, without therapeutic intervention, can lead either to an *exitus letalis* of the neonate within a short time post natum (p.n.) or to lasting developmental disorders.

The maturation of fetal pulmonary tissue and its state of maturity at the time of birth is highly species dependent. For instance, in altricial species, i.e., species in which the young are underdeveloped at the time of birth, including canine neonates, the lungs are morphologically and functionally immature, in a so-called canalicular–saccular stage. The postnatal onset of gas exchange in the lung tissue occurs via the thin-walled canaliculi and terminal sacculi [1,2]. Alveolar ducts and alveoli are completely formed postnatally [2,3,4]. Around 10 d. p.n., the canaliculi transform to alveolar ducts, and the sacculi develop to become alveoli, resulting in a morphologically and functionally intact gas exchange tissue [2].

However, the actual duration of this transformation in canine pulmonary tissue is still not known but seems to be breed-related and to depend on the individual birth weight and early postnatal physical development [5]. 

In dogs, the gestation period varies (63 ± 5 days); moreover, onset of birth has been shown to depend on the fetus’ number [6,7]. Bitches in which the number of fetuses exceeds the normal breed average by more than 20% will give birth earlier than those with a below-average litter size [7]. Physiologically, the extent to which the fetus number-dependent gestation period affects lung tissue development is still not known.

Moreover, whether (and, if so, at which timepoint) surfactant is formed in fetal canine lung tissue is unknown. In altricial species such as mouse and rat, it has been shown that minimal amounts of surfactant are synthesized by type 2 epithelial cells toward the end of the canalicular developmental period [8,9]; however, there are no such data available on fetal canine lung tissue.

These and other questions have clinical relevance regarding the handling of neonatal pups, especially when complications occur at the onset of respiratory function immediately p.n. In this study, using histopathological methods, we investigated lung aeration within the first adaptation period (0–24 h) to demonstrate the development of pulmonary tissue aeration within the first period of adaptation in vital born canine neonates using CT.

## 2. Animals, Materials, and Methods

### 2.1. Animals

A total of 40 canine neonates with a minimum birth weight between 230 and 610 g were included in the studies. Thirty-three of these came from three litters of two healthy dams (crossbreeds: Labrador: Retriever; Labrador: Boxer). The age of the dams at the time of birth was Ø 37.3 ±11.9 months, and the gestation period was 61.3 ± 2.05 days. All three births were free of complications (birth duration Ø 473.3 ± 143.8 min; inter-pup interval 43.03 ± 18.4 min) and were under continuous veterinary control. Clinical assessment of the neonatal vitality level from these three litters was performed immediately p.n. according to a modified Apgar score [5,10]. 

According to the modified Apgar score, five parameters were examined to investigate the vitality of the newborn pups. Heartrate, respiratory rate, color of mucus membranes, reflex irritability, and motility were checked. A detailed clinical examination of each pup from cranial to caudal took place. The signalment (identification, color, gender, weight) was recorded and documented. The upper airways were examined for signs of obstruction. The nostrils were inspected for foam blistering. This was followed by an inspection of the eyes and ears to ensure that the systems were completely developed. The oral cavity was examined for the presence of malformations (palatochisis). The color of the mucosa, the moisture, and the capillary refill time provided information about the peripheral blood flow. This was followed by checking the sucking and swallowing reflex. The lungs were auscultated in all four quadrants using a Littmann stethoscope to record breath sounds. To check the cardiovascular system, the apex beat was observed, the heart was auscultated, and a possible positive vein pulse of the vena jugularis was recorded. The abdomen was examined, and palpation and abdominal wall tension were recorded. The umbilicus was investigated for signs of dryness and inflammation, and it was palpated to rule out umbilical hernia. Finally, the correct attachment of the sexual organs, the tail, and the anus was verified, and the presence of an inguinal hernia was excluded. 

Each single vital born pup was immediately wrapped in a warmed blanket after the initial assessment (Apgar score) and scanned using computed tomography in abdomen–chest position. After that, a detailed clinical segmental control and determination of the birth weight (423.8 ± 80.1 g) were performed. In two pups from a litter free of complications (litter 3), measures had to be taken to stabilize the lung function due to a neonatal respiratory distress syndrome. This necessary resuscitation consisted of physiotherapeutic and medicinal procedures and was performed before CT examination [5]. The data for the 3 litters are shown in Table 1.

Litter 1 included 10 pups (8 female, 2 male), litter 2 included 12 pups (6 female, 6 male), and litter 3 included 11 pups (7 female, 4 male), of which 2 were stillborn, and 2 vital depressed (exitus 24 and 48 h p.n.). 

In addition, six other stillborn pups (total number of stillbirths: 8) and one other vital depressed neonate (exitus 30 min p.n.) from other litters of dams of the same body weight class (>20 kg) were included in the study for comparison purposes for the morphological imaging of lung tissue. Because no additional microbiological studies were performed on the pathogenesis of stillbirth, this limits the interpretation to some extent. This results in a total number of 40 animals.

The lungs of the 8 stillborn and the 3 neonatally depressed pups were examined histopathologically (*n* = 11). The tissue material was collected during routine obstetrics and neonatology procedures and was used with the consent of the owners.

The imaging analyses performed for the present study in the first adaptation period (0–24 h) took place with the consent of the owners and in the immediate temporal context of the birth. As the extended imaging examination procedure, in addition to routine clinical neonatal initial care, helped to ensure the survival of newborn pups, no animal experimentation application was necessary at the time of the investigations and no vote of the ethics committee was required. Under file number 7221.3-17493_18059_23, the competent authority (LALLF Mecklenburg-Vorpommern, Rostock, Gemany) certified that the project did not require approval and is not an animal experiment in the sense of Art. 7 para. 2 of the German Animal Welfare Act.

### 2.2. Imaging Technique

The computed tomography (CT) scanner used for the first litter was a dual slice spiral “Elscint Twin flash CT” (Elscint-GmbH, Wiesbaden, Germany), consisting of a scanner unit (gantry), positioning unit (couch), and control console. For the second and third litter, a different computed tomography scanner (Somatom Volume Zoom, Siemens, Germany) was used. However, there were no technical differences between the two devices that could influence the results.

Based on the clinical and practical conditions, the following groups were formed depending on the time of examination p.n.: Group 1: CT in the first 10 min p.n. (*n* = 11); Group 2: CT between 10 and 15 min p.n. (*n* = 15); Group 3: CT from 16 to 20 min p.n. (*n* = 5). This grouping resulted from the fact that the inter-pup intervals were sometimes very short (<30 min) and because delays occurred in the first measurement procedure because of this quick sequence of births. All these animals were re-examined using CT at 24 h p.n. (*n* = 31); CTs of stillborn infants (*n* = 8) served as controls for computed tomographic imaging of nonventilated lungs. All measurements of the thorax were taken with the pups in the chest–abdominal position. The slice thickness was uniformly 1.1 mm with an increment of 1. The pitch was 0.7. A high-resolution reconstruction matrix of 512 × 512 was used and the collected data were presented in grayscale. On average, the measurement time was 80–90 s and no longer than 120 s.

To cover all lung areas, the thoracic region was divided into four quadrants of equal size, with two dorsal (dors.) and two ventral (vent.) regions, left and right each. The principles of this methodology have been described earlier [11]. In each quadrant, three regions of interest (ROI) at the level of the second, fourth, sixth, and eighth intercostal spaces were used to measure the density of lung tissue using Hounsfield units (HU; positive values correspond to a higher X-ray density than the reference value for water, 0 HU). The corresponding arithmetic mean values were summarized for each of the dorsal and ventral quadrants and used for statistical analysis.

### 2.3. Histopathology and Morphometry

Lung tissue from all stillborn pups (*n* = 8) as well as two pups from litter 3 that died within 48 h p.n. and from another pup that died approximately 30 min after caesarean section (*n* = 3) was examined histopathologically, and the degree of ventilation was quantified histomorphometrically. For this purpose, lung tissue samples were fixed in 10% neutral buffered formalin, routinely embedded in paraffin wax, cut at 3 μm, and stained with hematoxylin and eosin for light microscopic examination. Transversal planes through the dorsal and ventral quadrants were used for histopathological evaluation.

A tissue classification algorithm was used for morphometry. This was trained by an American College of Veterinary Pathologists (ACVP) board-certified veterinary pathologist (J.P.T.) in an unblinded manner to detect tissue, edema, and optically empty areas (excluding bronchi and bronchioles) using HALO software version 3.1 (Indica Labs, Albuquerque, NM, USA). The tissue sections from all examined quadrants were then segmented using the trained algorithms to quantify the percentage of parenchyma and fluid vs. optically empty areas in the lung tissue. The percentage of optically empty areas (ventilated areas) was then compared to the total (solid) lung tissue.

### 2.4. Statistical Methods

Univariate statistical analysis of variance (ANOVA) with repeated measures regarding dorsal/ventral segmental (second, fourth, sixth, eighth intercostal space) ventilation levels and fixed effect in time groups (1: <10 min, 2: 10–15 min, 3: >15min) was performed using SAS 9.4 (SAS^®^ Institute Inc., 2013. Base SAS^®^ 9.4 Procedures Guide: Statistical Procedures, 2nd edition ed. Statistical Analysis System Institute Inc., Cary, NC, USA). A significance level of *p* < 0.05 was used.

## 3. Results

### 3.1. CT Images of Vital Pups Immediately p.n. and Stillborn Pups

Already, during the first 10 min p.n., a strong X-ray reduction in vital born pups was measured compared to stillborns, which served as a reference. The absorption reached mean values of −530 (±114) HU in dorsal and −453.3 (±133) HU in ventral lung quadrants, while, in stillborns, values of 35.80 (±18.17) HU in dorsal and 37.32 (±25.65) HU in ventral lung regions were measured, respectively.

CT images immediately p.n. of a stillborn puppy and of a puppy 30 min p.n. are shown in lung and soft tissue fenestration, respectively. The cross-section is selected in the region from the sixth intercostal space (Figure 1).

We further observed a tendency for ventilation to increase continuously within the first 20 min p.n. (Figure 2).

The dorsal lung areas were always significantly better (*p* = 0.0013) ventilated than the ventral regions (Figure 3). 

The results shown in Figure 2 and Figure 3 refer to the repeated measures’ ANOVA, in which only the results of the live-born animals were included, because, in the stillborn animals, the ventilation of the lungs could not be recorded at different time points.

The measured HU for the different groups are also shown (Figure 4). By re-examining all the animals at 24 h p.n., mean values of dorsal −648.0 (± 89.9) HU and ventral −624.7 (±76.8) HU were obtained.

In stillborn pups (*n* = 8) taken for CT examination immediately after birth, a strong absorption of X-rays was found in the lung parenchyma, which is in contrast to ventilated lung tissue (dorsally between +10 HU and +69 HU and ventrally between +18 HU and +70 HU). Only in one pup were negative values of −58 HU measured in the left dorsal and ventral quadrants, indicating that minimal amounts of air must at least have reached the sections of the left lung.

### 3.2. Histopathology of Stillborn Pups and of Puppies That Died Shortly p.n

In lung tissue specimens from stillborn pups, sacculi were consistently not unfolded and were judged to be primarily fetal atelectatic (Figure 5A and Figure 6).

For ethical reasons, no vital born, alive pups were euthanized for microscopic tissue examination. Instead, the lungs of pups that died between 0.5 and 48 h p.n. were used for comparison (*n* = 3). All of these neonates suffered from respiratory problems (neonatal respiratory distress syndrome, NRDS) from the beginning of the postnatal period. A histopathologic examination of their lungs revealed either low-grade ventilated or largely nonventilated (atelectatic) lung tissue except for bronchioles and bronchi. In some localizations, aspirated squames or other amniotic fluid components (e.g., meconium) were found, as well as exfoliated pneumocytes in terminal sacculi and bronchioles (Figure 6). However, some areas of the pulmonary tissue showed moderately aerated lung saccules and dilated bronchioles (Figure 7).

The optically empty area used as a measure for the degree of ventilation was significantly different (*p* = 0.0398) in stillborn pups (LSMeans estimate 13.9389, StdErr 2.8281, *n* = 8) compared to the vital born pups (LS Means estimate 26.9463, StdErr 4.6182, *n* = 3; Figure 8A). A histomorphometric examination of optically empty areas in the lung tissue of stillborn pups could not detect a difference between the dorsal and ventral lung areas (Figure 8B). 

## 4. Discussion

The aim of this study was to determine the time period after which a sufficient aeration of the lung tissue can be assumed. Furthermore, it should be determined whether there were differences in the uptake of function between the dorsal and the ventral lung segments. For comparison, stillborn pups were included in the first measurement period. 

The cause of stillbirth was not investigated in these cases and limits therefore interpretation as reference. It could be consequence of birth length, overlong inter-pup interval, or a sequela of an infection of the dam [12]. To this end, computed tomography was the applied method; moreover, if tissues were available, they were compared using histopathology. Over the past decades, computed tomography (CT) of the thorax has established itself as an important radiologic procedure. Computed tomographic imaging of the lung in a transverse plane provides unique diagnostic information that is unobtainable with conventional radiographic techniques. The pulmonary imaging of lung content can be evaluated, and CT images can therefore offer accurate information on pulmonary clearance during the immediate transition period of the neonate. In the future, electrical impedance tomography (EIT) will certainly become more widespread as a noninvasive, nonionizing, real-time imaging technique, and it is likely to become more important, including for canine neonatal pulmonology and neonatal intensive care [13]. However, to the best of our knowledge, no such suitable systems are currently available for canine neonates, since the accuracy of EIT is strongly related with a mesh corresponding to the different thoracic shapes of the dogs [14].

The results obtained using CT in canine neonates regarding the pulmonary tissue involved in gas exchange are primarily of scientific interest; however, they also have strong clinical relevance in the context of optimizing resuscitation procedures.

Approximately 75% of all perinatal losses in the canine species occur during or immediately after birth and thus in the first adaptation period (0–24 h p.n.) as well as in the initial period of the second adaptation period (second–fifth day of life). In Beagle dogs, 69.2% of all pup losses occurred in the first 72 h p.n. [12]. A major reason for this is that lung tissue in newborn dogs is still in a morphologically and functionally immature canalicular–saccular stage. During the fetal canalicular phase of lung development, the distal airways are formed, leading to the completion of bronchiolar branching. This is also the stage when surfactant is first detected in altricial species (e.g., mouse, rat, rhesus monkey, human), which appears to be related to the epithelial differentiation of the acini [9]. Reaching the canalicular stage is very important for very immature and premature neonates. This is because, at this time, the later air–blood barriers are first formed and, shortly after their appearance, type II epithelial cells begin to produce at least minimal amounts of surfactant. In most species, surfactant appears after about 80–90% of the gestational period, and even slightly earlier in humans. Small amounts of surfactant are then present after 60% of the gestational period (22–24 weeks) [15]. Initially, surfactant appears to be less abundant in the basal lung regions than in the apical, i.e., cranio-dorsal, regions in fetal lambs and rhesus monkeys [16,17].

From studies on lambs and rabbits, it is apparent that the lung areas mature at different rates, with the “upper” (i.e., cranio-dorsal) lobes dominating and the “lower” (caudo-ventral) lobes following thereafter [18,19]. In dogs, the cranial pulmonary lobes of the fetus are more developed than the caudal lobes, as well as in a centrifugal manner [20]. Lung alveolarization begins at the final stages of fetal development [21]. The increase in surfactant correlates temporally with the progressive maturation of the mechanical properties of the lung tissue [18]. Very high lecithin content is also found in neonatal rats for a short time after birth, gradually decreasing to adult levels from 3 to 5 days after birth, while, at the same time, favoring the greater dilatation of the terminal airways at the time of birth because of its biochemical and physical properties [8]. In canine pups, surfactant protein SP-B can be detected in lung parenchyma from as early as 55 days of gestation [21]. Only true alveoli are formed postnatally.

It is known that the first breaths of the canine neonate begin before the completion of expulsion to compensate for the no-longer-intact materno-placental gas exchange *sub partu* [5]. The thoracic compression caused by the narrowness of the soft birthway is released when the head and thoracic segments have passed the *rima vulvae* and the first breaths occur, leading to an expansion of the thorax, inflation of the pulmonary alveoli, and, henceforth, regular inspiration and expiration [5]. As a result of the inflation, the cubic epithelial cell association of the bronchiolar ducts is ruptured into metameric cell clusters between which alveolar sacs with alveoli are immediately formed [2]. As a special feature of canine neonates, their lungs can remain in an apneic inspiratory position for up to 20 s after the inflow of the first air stream [5]. This apparently causes more distant alveoli in the terminal tip lobes of the lung to dilate, allowing more intensive gas to exchange before expiration begins.

Yet another circumstance should be noted regarding the development of the first pulmonary gas exchange in canine neonates. Due to the variable gestation period, it can be hypothesized that the lung tissue has a different developmental state immediately post natum.

It was found that, during the first 10 min after birth, in canine neonates, about 73% of the lung tissue is aerated. Up to 20 min p.n., ventilation increases further, so that 82% of the pulmonary area is already ventilated. The maximum is reached at 24 h p.n. These findings could be important in the resuscitation of neonates with neonatal respiratory depression. Thus, it is important that appropriate targeted physical resuscitation procedures are applied in the first 10 min p.n. to ventilate as much of the lung tissue as quickly as possible. Because of the differences in the degree of ventilation described here, resuscitation with the pup in the thoracic position seems more appropriate. It can be assumed that this recommendation is particularly relevant in neonates who developed through obstetric laparotomy. In the latter, in contrast to pups born per *vias naturalis*, the particular complication is that the fluid present in the bronchial tree has not been squeezed off, which normally occurs via thoracic compression as the pup slides through the bony and soft birth canal. Thus, from a clinical point of view, regarding the sequence and intensity in the course of resuscitation procedures within the first 10 min p.n., it would be necessary to consider whether the neonate was born per *vias naturalis* or by obstetric laparotomy [5]. It has also been demonstrated that, in this species, the gestation length depends on the litter size [6,7]. Bitches in which the number of fetuses is above the normal breed average give birth earlier than those with a below-average number of fetuses, in which duration of pregnancy is prolonged. Therefore, in Boxer bitches, the gestation period varies considerably (55 to 71 days). Conversely, it can be assumed that the lung tissue also shows a different developmental state depending on the gestation period.

The pathomorphological equivalent of incomplete lung function is atelectasis. This is understood to be an incomplete expansion of the terminal air-bearing pathways, which is present, for example, as congenital atelectasis when the lungs have not filled sufficiently with air after birth.

The main intrauterine function of the lungs is considered to be the production of fetal lung fluid, which drains into the amniotic cavity via the trachea, oral cavity, and nasal cavity or is swallowed [22]. In lambs, surfactant has been detected in the amniotic fluid [23]. At birth, some of the fetal lung fluid is forced out of the bronchial tree during thoracic compression as the fetus passes through the maternal pelvis and vagina and is either swallowed by the neonate or becomes lodged in and obstructs the upper airway [4]. The other part of the fluid is rapidly reabsorbed by the changing blood pressure conditions in the small circulation due to adrenaline and vasopressin release and is drained via lymphatic and capillary channels [4]. This results in the successive unfolding of the terminal sacculi. Congenital atelectasis thus occurs in neonates which are unable to inflate their lungs after the first few breaths. It may also be caused by airway obstruction, often as a result of the aspiration of amniotic fluid and meconium (meconium aspiration syndrome). Congenital atelectasis also occurs when the alveoli cannot remain inflated after the first intake of air due to the qualitatively and quantitatively insufficient production of antiatelectase factor (surface active agent, surfactant) produced by type II pneumocytes and Clara cells, and instead collapse [24]. This form of congenital or neonatal atelectasis (also dystelectasis) is called acute respiratory distress syndrome in human neonatology [5]. Reportedly, the amount of surfactant varies greatly from individual to individual, which in turn strongly influences the postnatal initiation of the gas exchange process in a state of deficiency [2,25].

## 5. Conclusions

In conclusion, the lung tissue of canine neonates presents an aeration profile as early as 10 min after birth and continues progressively, with a special regard to the dorsal lung areas. These results might have clinical relevance in that, particularly in the context of necessary resuscitation measures, they suggest that the procedure should be performed with the pup in the abdomen–chest position to achieve the highest possible degree of ventilated pulmonary tissue. The investigations carried out here have provided a basis for evaluating the most appropriate resuscitation measures in further studies.

## Figures and Tables

**Figure 1 animals-13-01741-f001:**
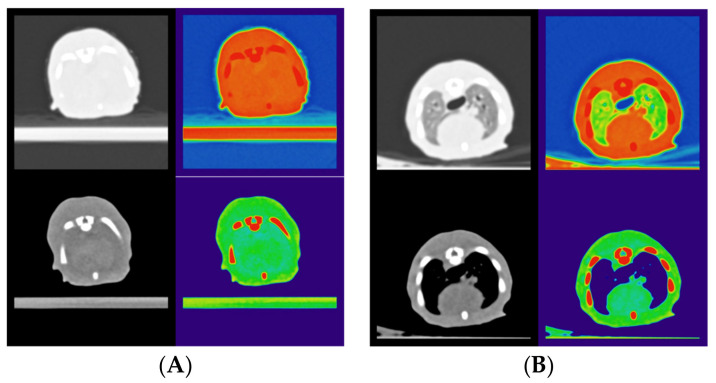
CT images immediately p.n. of a stillborn puppy (**A**) and of a viable puppy 30 min p.n. (**B**) are shown in lung and soft tissue fenestration, respectively. The cross-section plane is selected in the region from the sixth intercostal space. Color-coded intensity images were produced using Fiji/Image (ImageJ 2.9.0/1.53t).

**Figure 2 animals-13-01741-f002:**
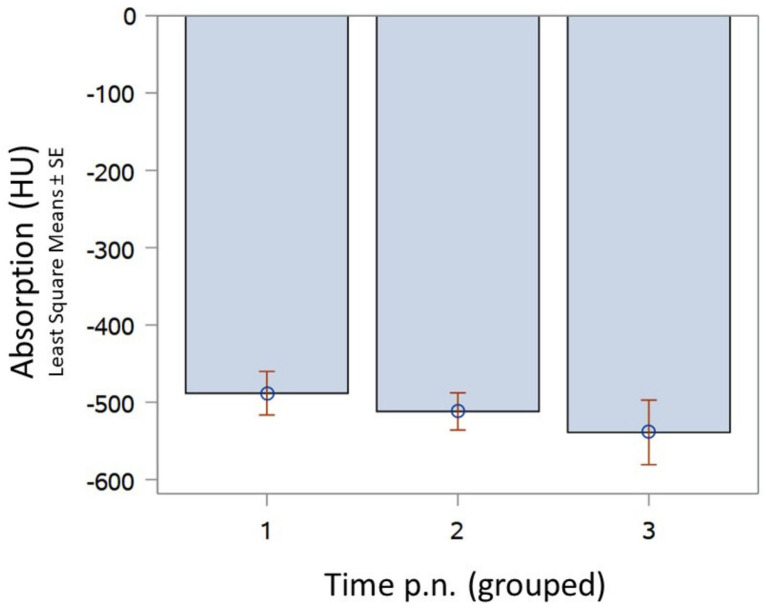
Comparison of CT readings from lung tissue immediately p.n. (groups I–III) in vital born pups. Time (p.n.) group 1: 0–10 min p.n. (*n* = 11) estimate −488.36; StdErr 28.1407; group 2: >10–15 min p.n. (*n* = 15) estimate −511.87; StdErr 24.0983; group 3: >15–20 min p.n. (*n* = 5) estimate −538.99; StdErr 41.7394.

**Figure 3 animals-13-01741-f003:**
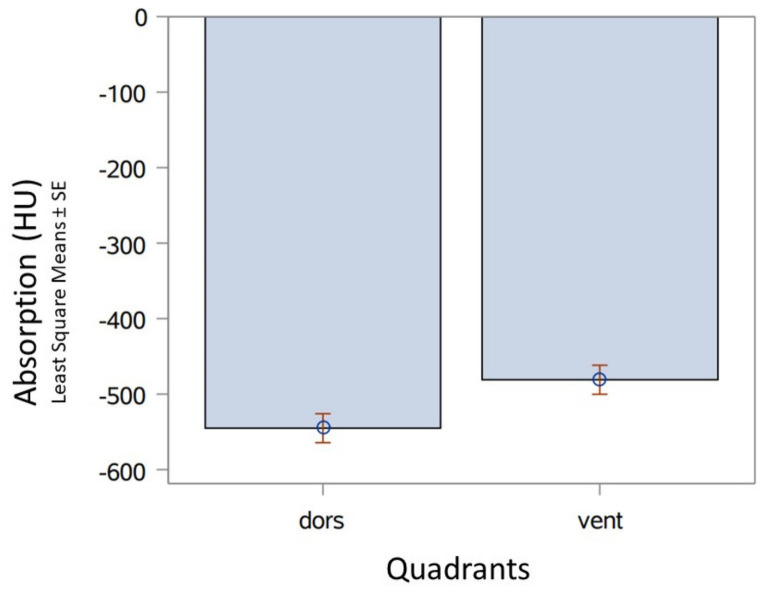
Comparison of Least Square Means (estimates) of CT dorsal (dors.; estimate −545.17; StdErr 19.2097) with ventral (vent.; estimate −480.97; StdErr 19.2097) quadrants in vital born pups directly post natum (time p.n. groups 1, 2, 3; *n* = 31).

**Figure 4 animals-13-01741-f004:**
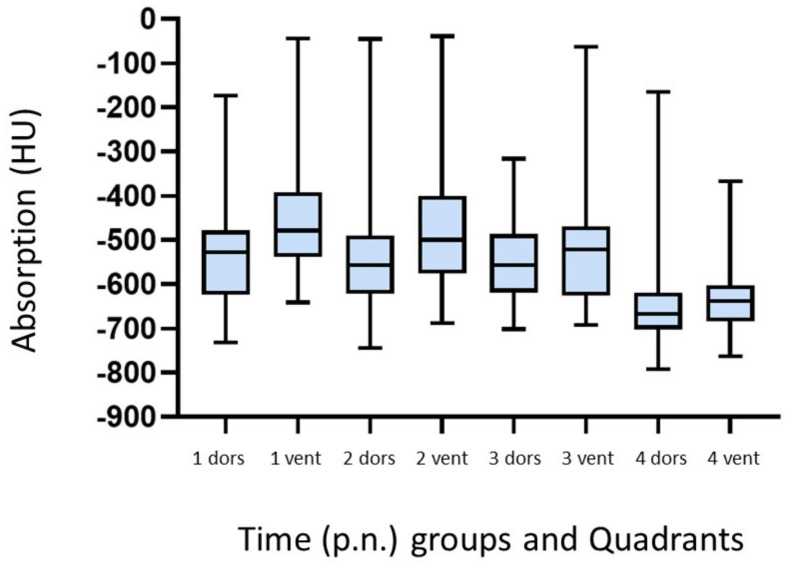
Comparison of measured HU for dorsal and ventral quadrants in the different time (p.n.) groups and 24 h p.n. of vital born pups (*n* = 31). Group 1: 0–10 min p.n. (*n* = 11); group 2: >10–15 min p.n. (*n* = 15); group 3: >15–20 min p.n. (*n* = 5); group 4: re-examination of all animals at 24 h p.n. (*n* = 31).

**Figure 5 animals-13-01741-f005:**
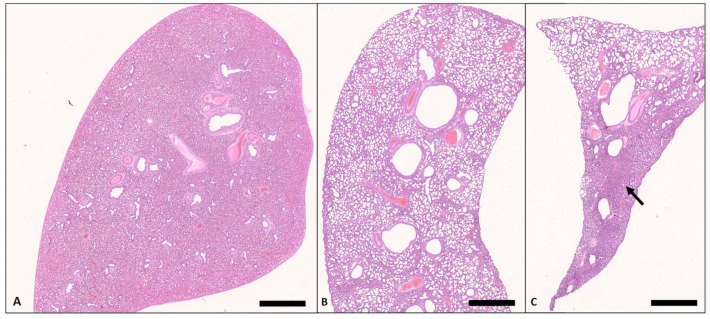
Lung, pups, H.E. (**A**) Dorsal lobe, stillborn, there is diffuse neonatal atelectasia (Bar = 3 mm). (**B**) Dorsal lobe, death 48 h p.n. Except for marginal areas, the lung tissue is diffusely ventilated (Bar = 1.25 mm). (**C**) Ventral lobe, death 48 h p.n.; there are abundant areas (arrow) of neonatal atelectasia (Bar = 250 μm).

**Figure 6 animals-13-01741-f006:**
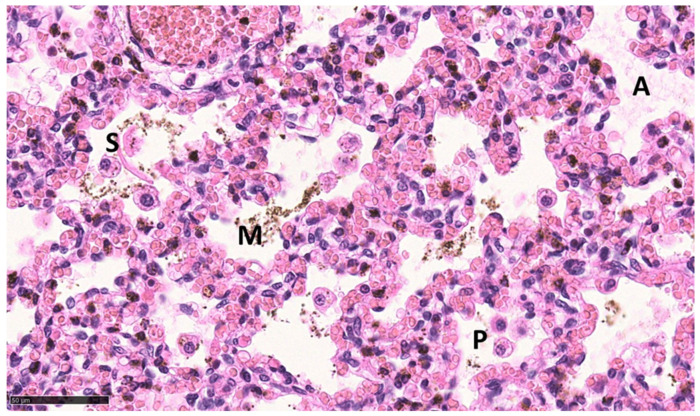
Lung, dorsal lobe, pup, stillborn, H.E. Fetal lung tissue with airways partly filled with amniotic fluid (A), exfoliated pneumocytes (P), meconium (M), and squames (S); Bar: 50 µm.

**Figure 7 animals-13-01741-f007:**
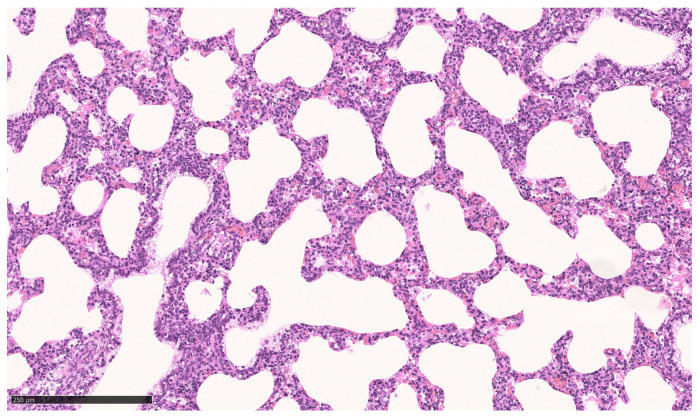
Lung, dorsal lobe of vital pup after death 48 h p.n. Lung saccules and bronchioles are moderately dilated; Bar: 250 µm.

**Figure 8 animals-13-01741-f008:**
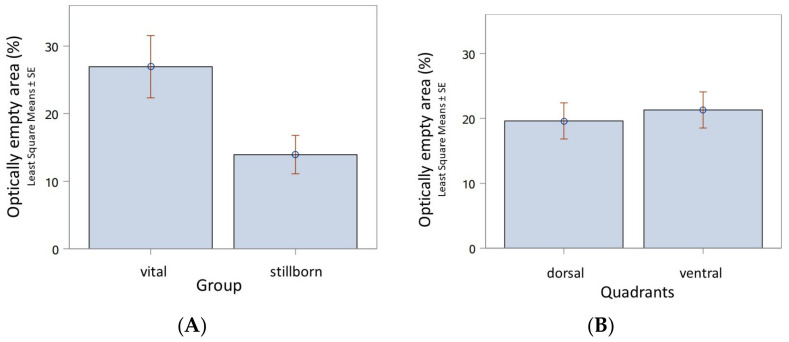
Histomorphometric analysis of the degree of the ventilation of the lungs in vital born and postnatally deceased canine pups (*n* = 3) and stillborns (*n* = 8). (**A**) The optically empty area in the lung tissue, when used as a measure of ventilation, is significantly different between live born pups and stillborn pups (*p* = 0.0398). (**B**) There are no anatomically based differences in stillborn pups with respect to the optically empty area in the lung tissue of dorsal and ventral regions (*p* = 0.2353).

**Table 1 animals-13-01741-t001:** Animal collective for CT examinations and histopathology. Both neonates died between 24 and 48 h p.n.

Litter #	Litter Size	Neonatal Status			♀	♂
		Vital	Vital Depressive	Stillborn		
1	10	10	0	0	8	2
2	12	12	0	0	6	6
3	11	7	2	2	7	4
sum	33	29	2 *	2 *	21	12
Add-on	-	-	1 *^†^	6 *	-	-
Total	*n* = 40					

* From these 11 animals, tissues were obtained for histopathology. ^†^ Caesarean section.

## Data Availability

The datasets generated and analyzed during the current study are available from the corresponding author upon request.

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
