# Peer review of "Computed Tomographic and Histopathologic Studies of Lung Function Immediately Post Natum in Canine Neonates"

_animals, 2023, doi:10.3390/ani13111741_

Round 1

Reviewer 1 Report

In their manuscript "Computed tomographic and histomorphometric studies of lung function immediately post natum in canine neonates" the Authors try to describe the development of lungs in dogs in short period after birth.

Although the topic is important and can influence the safety of canine neonates, in my opinion the study has serious flaws that do not allow a publication in the presented form.

In the aims of their study Authors mention the description of histopathological image of the lungs in dogs dying during the first 24h of life (aim number 2); nonetheless, the number of dogs examined histologically and compared to stillborn pups is very small (n=3) and therefore in my opinion do not allow to draw any significant conclusions, as the possibility of error is relatively high. I understand that obtaining a higher number of dogs dying shortly after birth can be challenging, but it do not dispense us from publishing valuable research and results. Therefore, either the histological and morphometric study should not be mentioned as an aim of the study (and in the title of the paper), or the number of examined animals should be increased.

Secondly, in lines 117-118 Authors mention that the CT examination was a part of extended examination procedure and therefore did not require the approval of ethics committee and that the imaging examination helped to ensure the survival of the pups. I do not understand how the examination could help in the survival of animals? Is the double CT examination (directly after birth and after 24 h) used in the clinic, where the study was performed, as a routine practice in newborn dogs? If not, in my opinion, at least a statement from the local ethical committee should have been obtained to avoid misdoubts. If the Authors' standpoint is based on the state law (as it can differ between the countries), please indicate that in the text with appropriate reference.

Next, the study was conducted using two various CT devices. The Authors do not clarify if there were significant differences in the two devices that could influence the results. This should be either clarified or mentioned in the limitation section.

Moreover, the causes of animals' death prior or after birth are not recognised and therefore should be mentioned as a study limitation, as various factors (mainly infectious agents) can influence the development of the respiratory system during pregnancy.

Hereafter I list also the minor revisions that should be addressed:

- line 33: it is not clear what the abbreviation "p.n" states for in this sentence? Did the Authors mean "3 early post natum neonates"?

- throughout the text: please change reference format from [1], [2] to [1,2], etc.

- lines 68-69: This sentence is not clear - please rephrase

- lines 96-100: was the CT examination in those two pups performed before or after the resuscitation? Or was it not performed at all? It is not clear.

- lines 126 and further: what was the duration of the CT examination?

- lines 126-136: the group design is not clearly presented; I believe that the animals (n=31) were divided into three groups and then re-examined after 24 hours but it is not described properly; basing on the information provided by the Authors the study was conducted on 62 animals (G1 n=11, G2 n=15, G3 n=5, G4 n=31)

- line 170: in my opinion the values obtained in the stillborn pups (reference) should be also presented in the Figure 1

- lines 169-194: please consider combining section 3.1 and 3.2 as the results obtained in stillborn pups are used as a reference for alive ones and are mentioned in the section 3.1

- lines 177-178: it is not clear in which time point were this results taken: does it refer to measurements obtained in ≤10 min, 10-15 min and 16-20 min for group 1, 2 and 3, respectively (if yes, it should be clearly stated and the results should be noted for each of the groups) or to the measurements obtained 24h after birth?

- Figure 1: are the measurements taken 24h after birth presented in the figure? It is not clear for me, as the figure present group 1, group 2 and group 3

- Figure 2: as mentioned previously - what was the time of those measurements?

- lines 195-212: as previously, please consider combining section 3.3 and 3.4 as one

- line 208: please explain the abbreviation "NANS"

- lines 211-212: this description reflects the histopathological image observed in stillborn pups and presented in Figure 5 - please clarify

- line 212, parenthesis: please correct to Figure 4C

- Figure 3: please add a reference to Fig. 3 in the text; please clarify if the Fig. 3 present the same data as Fig. 2 but in each studied group? Was there a difference in the results observed between the groups?

- Figure 4, legend: in the description of C please rephrase to "C) Ventral lobe, death 48h p.n...." to keep concise with the rest of figure legend

- Figure 6: there is no reference to Fig 6 in the manuscript - please add

- Figure 7: there is no reference to Fig. 7A in the manuscript - please add; Figure 7B do not correspond with the legend of Figure 7 - please either rephrase the legend of Fig. 7 to include the difference in dorsal vs. ventral lobes in stillborn pups or divide Fig. 7 to two separate figures.

- lines 282-292: I appreciate that the mentioned facts "are known" to the Authors, but still some references confirming those statements would be useful; otherwise it is just a hypothesis and not a discussion with the literature

- lines 294-296: as above - please add references to that statement

- lines 298-299: Group 3 was examined up to 20 minutes after birth; what is the basis of the statement that 82% of the lungs is inflated up to 30 minutes p.n.?

- line 299: the parenthesis is unnecessary

- lines 300-301: please rephrase this sentence

- lines 300-306: the conclusion drown in this paragraph is not confirmed by the conducted study; the Authors did not perform studies on the impact of the time of resuscitation on the lungs inflation, they only stated that the lungs inflate in approx. 73% during the first 10 minutes after birth (which can be identified with the first breath). But the Authors only examined the inflation in normal lungs in healthy puppies. The respiratory depression can influence the lung capacity and duration of full inflation, nonetheless also a situation is possible when the full inflation will be obtained even if the first breath is forced few minutes after birth. It should be examined in controlled manner and only then such conclusions can be drawn

- line 306: as above: what is the basis of limitation of 30 minutes of resuscitation; the Authors did not examine the impact of starting point and duration of resuscitation on the pulmonary capacity and inflation

- line 312-313: this statement is not supported by the results of the study (only one puppy was delivered by CC in the study)

- lines 313-318: I find it difficult to follow whether this paragraph is related to the impact of the number of foetuses on the duration of pregnancy or on the way of birth (natural or CC)

- lines 325: what is the relation of the presence of surfactant in the lambs' lungs to the rest of the paragraph? I find it unclear

- lines 326-342: please add references to parts of this paragraph (and not only at the end)

- lines 343-347: according to previously mentioned doubts the conclusions are unfounded and exaggerated; basing on their study, the Authors cannot draw conclusions on the efficacy of resuscitation depending on the time of its initiation and duration

- References: almost half of the cited references (11 out of 23) are older than 1990 and three further are from 1990's. In my opinion newer references should be discussed to show the importance of the research to 21st century veterinary science

Author Response

REVIEWER #1

Responses

1.      

In their manuscript "Computed tomographic and histomorphometric studies of lung function immediately post natum in canine neonates" the Authors try to describe the development of lungs in dogs in short period after birth.

/

2.      

Although the topic is important and can influence the safety of canine neonates, in my opinion the study has serious flaws that do not allow a publication in the presented form.

/

3.      

In the aims of their study Authors mention the description of histopathological image of the lungs in dogs dying during the first 24h of life (aim number 2); nonetheless, the number of dogs examined histologically and compared to stillborn pups is very small (n=3) and therefore in my opinion do not allow to draw any significant conclusions, as the possibility of error is relatively high. I understand that obtaining a higher number of dogs dying shortly after birth can be challenging, but it do not dispense us from publishing valuable research and results. Therefore, either the histological and morphometric study should not be mentioned as an aim of the study (and in the title of the paper), or the number of examined animals should be increased.

- Title: “histomorphometric” was replaced by “histopathologic”

- Aim: #2 was deleted and histopathology as method integrated in the aim (#1): “...to demonstrate the development of pulmonary tissue ventilation aeration within the first period of adaptation in vital born canine neonates by CT, and using also histopathological methods.

Since histopathology is an integral and important part of this study we do not agree to skip it completely from the title, abstract or aim.

4.      

Secondly, in lines 117-118 Authors mention that the CT examination was a part of extended examination procedure and therefore did not require the approval of ethics committee and that the imaging examination helped to ensure the survival of the pups. I do not understand how the examination could help in the survival of animals? Is the double CT examination (directly after birth and after 24 h) used in the clinic, where the study was performed, as a routine practice in newborn dogs? If not, in my opinion, at least a statement from the local ethical committee should have been obtained to avoid misdoubts. If the Authors' standpoint is based on the state law (as it can differ between the countries), please indicate that in the text with appropriate reference.

Under file number 7221.3-2.039/13, the competent authority (LALLF Mecklenburg-Vorpommern, Rostock) certified that the project did not require approval under Section 8a (1) (2) of the Animal Protection Act (in 2013 when the first experiments started).

This was added lines 130 ff.

5.      

Next, the study was conducted using two various CT devices. The Authors do not clarify if there were significant differences in the two devices that could influence the results. This should be either clarified or mentioned in the limitation section.

line 138: This was clarified by “However, there were no technical differences in the two devices that could influence the results.”

6.      

Moreover, the causes of animals' death prior or after birth are not recognised and therefore should be mentioned as a study limitation, as various factors (mainly infectious agents) can influence the development of the respiratory system during pregnancy.

It was not the aim of the study to determine the causes of death in stillbirths - all stillbirths were so-called newborn pups that died during birth. This may be a consequence of birth length, overlong inter-pup interval. The specific causes could not be determined clinically. There were no indications of infectious causes or malformations.

7.      

Hereafter I list also the minor revisions that should be addressed:

/

8.      

- line 33: it is not clear what the abbreviation "p.n" states for in this sentence? Did the Authors mean "3 early post natum neonates"?

Was rephrased: “(8 stillborn and 3 neonates, died early post natum, respectively)”

9.      

- throughout the text: please change reference format from [1], [2] to [1,2], etc.

Was done accordingly

10.   

- lines 68-69: This sentence is not clear - please rephrase

Was rephrased to: “Bitches in which the number of fetuses exceeds the normal breed average by more than 20% will give birth earlier than those with a below average litter size.”

11.   

- lines 96-100: was the CT examination in those two pups performed before or after the resuscitation? Or was it not performed at all? It is not clear.

line 107: rephrased “This necessary resuscitation consisted of physiotherapeutic and medicinal procedures and was performed before CT examination [5].”

12.   

- lines 126 and further: what was the duration of the CT examination?

CT took 80-90 sec (at max. 120 sec.) – was added in lines150-151

13.   

- lines 126-136: the group design is not clearly presented; I believe that the animals (n=31) were divided into three groups and then re-examined after 24 hours but it is not described properly; basing on the information provided by the Authors the study was conducted on 62 animals (G1 n=11, G2 n=15, G3 n=5, G4 n=31)

This was rephrased and “group 4” more clearly described as: “All these animals were re-examined by CT at 24 h p.n.”

14.   

- line 170: in my opinion the values obtained in the stillborn pups (reference) should be also presented in the Figure 1

The results shown in Figures 2 and 3 refer to the repeated measures ANOVA (thus estimates), in which only the results of the live-born animals were included, because in the stillborn animals the ventilation of the lungs could not be recorded at different time points. (this explanation was also added in lines 201ff)

15.   

- lines 169-194: please consider combining section 3.1 and 3.2 as the results obtained in stillborn pups are used as a reference for alive ones and are mentioned in the section 3.1

Was done as recommended

16.   

- lines 177-178: it is not clear in which time point were this results taken: does it refer to measurements obtained in ≤10 min, 10-15 min and 16-20 min for group 1, 2 and 3, respectively (if yes, it should be clearly stated and the results should be noted for each of the groups) or to the measurements obtained 24h after birth?

17.   

- Figure 1: are the measurements taken 24h after birth presented in the figure? It is not clear for me, as the figure present group 1, group 2 and group 3

See #16.

18.   

- Figure 2: as mentioned previously - what was the time of those measurements?

It was clarified by: “directly post natum (time p.n. groups 1,2,3; n=31)

The results shown in Figure 3 refer also to the repeated measures ANOVA (thus estimates), in which only the results of the live-born animals were included, because in the stillborn animals the ventilation of the lungs could not be recorded at different time points.

19.   

- lines 195-212: as previously, please consider combining section 3.3 and 3.4 as one

Was done as recommended

20.   

- line 208: please explain the abbreviation "NANS"

was replaced by “NRDS”

21.   

- lines 211-212: this description reflects the histopathological image observed in stillborn pups and presented in Figure 5 - please clarify

the description actually refers to figure 6; “3C” remained from an earlier version of the manuscript and is incorrect, was replaced

22.   

- line 212, parenthesis: please correct to Figure 4C

3C was replaced by 6

23.   

- Figure 3: please add a reference to Fig. 3 in the text; please clarify if the Fig. 3 present the same data as Fig. 2 but in each studied group? Was there a difference in the results observed between the groups?

These are the “measured HU” and not the estimated values and therefore not identic to figure 3

“The measured HU are depicted in Figure 4.” Was added in line 204

24.   

- Figure 4, legend: in the description of C please rephrase to "C) Ventral lobe, death 48h p.n...." to keep concise with the rest of figure legend

Was done accordingly

25.   

- Figure 6: there is no reference to Fig 6 in the manuscript - please add

Ref. was added in lines 217-218: However, some areas of the pulmonary tissue showed moderately aerated lung saccules and dilated bronchioles (Figure 6).

26.   

- Figure 7: there is no reference to Fig. 7A in the manuscript - please add; Figure 7B do not correspond with the legend of Figure 7 - please either rephrase the legend of Fig. 7 to include the difference in dorsal vs. ventral lobes in stillborn pups or divide Fig. 7 to two separate figures.

Reference Figure 8A, 8B was added in line 221-224

The legend for 8B. is correct as it stands and corresponds to the figure for the stillborn pups: “There are no anatomically based differences in stillborn pups with respect to optically empty area in lung tissue of dorsal and ventral regions (P = 0.2353).”

27.   

- lines 282-292: I appreciate that the mentioned facts "are known" to the Authors, but still some references confirming those statements would be useful; otherwise it is just a hypothesis and not a discussion with the literature

Potentially the line numbering has changed due to format issues? This comment is not specific enough since as it stands, since in lines 282-292 we cite [9,15,16,17] which we consider sufficient. If we missed the point of the reviewer here, please specify the “facts” and check line numbers again.

28.   

- lines 294-296: as above - please add references to that statement

Again, perhaps wrong line numbers? For 294-296 we cited [8]. Or specify, which statement does the reviewer mean in detail.

29.   

- lines 298-299: Group 3 was examined up to 20 minutes after birth; what is the basis of the statement that 82% of the lungs is inflated up to 30 minutes p.n.?

Was changed to 20 min; Howevr, 30 min result from long clinical experiences for reanimation of neonatal canines

30.   

- line 299: the parenthesis is unnecessary

The line numbers seem to be in disarray, please specify which parenthesis is meant

31.   

- lines 300-301: please rephrase this sentence

If lines 320-321 are meant? “It is these findings that would need to be considered when resuscitating canine neonates suffering from neonatal respiratory depression” This was rephrased to “These findings need to be considered when resuscitating neonates with neonatal respiratory depression.”

32.   

- lines 300-306: the conclusion drown in this paragraph is not confirmed by the conducted study; the Authors did not perform studies on the impact of the time of resuscitation on the lungs inflation, they only stated that the lungs inflate in approx. 73% during the first 10 minutes after birth (which can be identified with the first breath). But the Authors only examined the inflation in normal lungs in healthy puppies. The respiratory depression can influence the lung capacity and duration of full inflation, nonetheless also a situation is possible when the full inflation will be obtained even if the first breath is forced few minutes after birth. It should be examined in controlled manner and only then such conclusions can be drawn

Are lines 318 meant?

The studies have shown that the first 10 min are the most important for lung development. Our investigations carried out have provided the physiological basis for studying the most appropriate resuscitation measures in further studies. Further investigation is undoubtedly important but not the focus of this study.

33.   

- line 306: as above: what is the basis of limitation of 30 minutes of resuscitation; the Authors did not examine the impact of starting point and duration of resuscitation on the pulmonary capacity and inflation

this was not the aim of the work to optimize the resuscitation methods - completely different question

34.   

- line 312-313: this statement is not supported by the results of the study (only one puppy was delivered by CC in the study)

lines 327-328? Was rephrased to: “It can be assumed that this recommendation is particularly relevant in neonates who developed through obstetric laparotomy.”

The situation of lung development in CC pups would have to be controlled in another study, which does not get its lung fluid squeezed out per vias naturales. This would also be a very extensive further study which is not the aim of this work. - it is a crucial hint that is gratefully received but cannot be implemented in this manuscript.

35.   

- lines 313-318: I find it difficult to follow whether this paragraph is related to the impact of the number of foetuses on the duration of pregnancy or on the way of birth (natural or CC)

line 358: it is highly significant and results from literature – has nothing to do with way of birth only length of gestation – questionable if longer gestation results in better development of the lungs

36.   

- lines 325 (345?): what is the relation of the presence of surfactant in the lambs' lungs to the rest of the paragraph? I find it unclear

In canine pups, surfactant protein SP-B can be detected in the lung parenchyma as early as 55 days of gestation [21].

37.   

- lines 326-342: please add references to parts of this paragraph (and not only at the end)

Which lines are meant? Please refer to specific sentences

In 326-342 the refs [6,7] were cited which cover the complete paragraph

38.   

- lines 343-347: according to previously mentioned doubts the conclusions are unfounded and exaggerated; basing on their study, the Authors cannot draw conclusions on the efficacy of resuscitation depending on the time of its initiation and duration

Which lines are meant? Please refer to specific sentences

39.   

- References: almost half of the cited references (11 out of 23) are older than 1990 and three further are from 1990's. In my opinion newer references should be discussed to show the importance of the research to 21st century veterinary science

We have now cited Regazzi et al. and Sipriani et al. as recent papers; however, even these unfortunately only cite older literature because not much new exists

Sipriani, T. M.; Grandi, F.; da Silva, L. C.; Maiorka, P. C.; Vannucchi, C. I. Pulmonary maturation in canine foetuses from early pregnancy to parturition. Reprod Domest Anim. 2009, 44 Suppl 2, 137–140.

Regazzi, F. M.; Silva, L. C. G.; Lúcio, C. F.; Veiga, G. A. L.; Angrimani, D. S. R.; Vannucchi, C. I. Morphometric and func-tional pulmonary changes of premature neonatal puppies after antenatal corticoid therapy. Theriogenology. 2020, 153, 19–26.

Reviewer 2 Report

The manuscript is well structured with the specifications spelled out throughout.

Excellent use of innovative technologies as a choice for a difference in results

Author Response

REVIEWER #2

1.      

The manuscript is well structured with the specifications spelled out throughout.

no action needed

2.      

Excellent use of innovative technologies as a choice for a difference in results

no action needed

Reviewer 3 Report

The present manuscript aims to characterize morphological features of the pulmonary parenchyma of neonatal puppies throughout the first 30 minutes after birth, based on thoracic tomography, taking into account stillbirths or subsequent morbi-mortality. For this purpose, neonates were classified as live and stillborn puppies. The portion of the lung analyzed by tomography was also considered.

This manuscript is interesting and has a novelty of analyzing lung structural development by a more accurate imaging technique. The study on puppies’ assessment as a clinical parameter in neonatology should be encouraged. Despite the interesting area of experimentation, there are important issues that derail its publication in the present form. The material and methods section lacks some essential information.

My most important concern relates to the discussion itself. Perhaps it would better to focus your discussion on pulmonary clearance and lung content that can impair neonatal respiratory function.

Some of the major comments and criticisms include:

Title:

I understand the meaning of “histomorphometric” analysis of the lung. However, morphometry encompasses a quantitative study of lung histologic architecture. Authors have not performed a quantitative analysis, but a descriptive histological evaluation of the lung components. Please, revise the title terminology

Simple Summary

Line 15: please clarify the meaning of “ventilation”. One can interpret as artificial lung insufflation other than natural pulmonary expansion.

Abstract

Line 37-38: Your conclusion is not paired with your objectives. Please, reformulate the conclusion in order to answer the scientific question established at the objective. In addition, conclusion is too overstated based on what authors have really performed in their methodology.

Introduction:

Although the Introduction is well written, authors should try to address thoroughly the respiratory physiology of the neonatal period.

Line 57 and 66-68: these statements need a reference for canine neonates

Lines 60-61: New data from Regazzi et al Morphometric and functional pulmonary changes of premature neonatal puppies after antenatal corticoid therapy (2020) show that lung alveolarization starts at the final stages of fetal development. Thus, alveoli can be found in recent born puppies.

Lines 70-71: Again, results from Regazzi et al. (2020) show that surfactant protein SP-B can be detected in lung parenchyma early as 55 days of gestation.

Line 75: above the lung functioning during the neonatal transition period, pulmonary clearance is of utmost importance to achieve full pulmonary gas exchange. 

Line 78: how exactly did authors perform the evaluation of gas exchange capacity? By reading the methodology, the primary postnatal function of the respiratory system to provide sufficient oxygenation and to eliminate CO2 was not analyzed in the present manuscript. Please, clarify

Line 79: see comment above for the term “ventilation”

Material and methods:

Some very important information should be provided:

-        How exactly the gestation period was calculated? Based on ovulation or LH peak?

-        Were all births vaginal deliveries?

-        What was the cause of death of stillbirth puppies? Was a necropsy performed?

-        The statistical analysis is too shallow. Please, include the probability values used to consider significance (P).

Line 98: how the respiratory distress syndrome was diagnosed?

Line 99: please detail the medical procedures executed

Line 104: the inclusion of a surgical born puppy biases the work, as recently shown by Almeida et al. Both spontaneous vaginal delivery and elective caesarean section influence neonatal redox status in dogs (2022)

Line 110: I am not sure that using the term “control” is the correct nomenclature for this situation. Perhaps it will be better to name it as “non-insufflated control” or “negative control”

Lines 117-118: I do not think that ethical approve has to be solely for experimental animals. The consent of an ethical committee has to be obtained regardless

Results:
Line 188: Perhaps it would be illustrative to include the CT comparative images.

Line 212: Figure 4C

Discussion

Line 237-238: if a functional analysis of blood oxygenation was not performed, you cannot affirm that gas exchange is being fully achieved.

Line 240-241: the use of stillborn as control animals is a matter of doubt. Is it possible to affirm that fetal death occurred during whelping? Can you exclude premature fetal death? Thus, the atelectasic aspect of the lung may be only a consequence of immature lung development in a canalicular phase (please see Sipriani et al., Pulmonary maturation in canine foetuses from early pregnancy to parturition 2009)

Lines 255-257: this statement needs reference

Line 259: saccular

Line 272-274: According to the study of Sipriani et al. (2009), cranial pulmonary lobes of the canine fetus are more developed than caudal lobes, as well as in a centrifugal manner.

Lines 279-281: please refer to the manuscripts of Sipriani et al. (2009) and Regazzi et al. (2020).

Lines 289-291: this statement needs reference

Line 298: perhaps it would better to say "inflate or insufflate" instead of “gas exchange”

Conclusion

The conclusion should be mirrored by the objectives. Although very useful, future application of the data should be stated, but it cannot be considered the conclusion of the research. Please, rewrite.

Author Response

REVIEWER #3

1.      

The present manuscript aims to characterize morphological features of the pulmonary parenchyma of neonatal puppies throughout the first 30 minutes after birth, based on thoracic tomography, taking into account stillbirths or subsequent morbi-mortality. For this purpose, neonates were classified as live and stillborn puppies. The portion of the lung analyzed by tomography was also considered.

/

2.      

This manuscript is interesting and has a novelty of analyzing lung structural development by a more accurate imaging technique. The study on puppies’ assessment as a clinical parameter in neonatology should be encouraged. Despite the interesting area of experimentation, there are important issues that derail its publication in the present form. The material and methods section lacks some essential information.

/

3.      

My most important concern relates to the discussion itself. Perhaps it would better to focus your discussion on pulmonary clearance and lung content that can impair neonatal respiratory function.

The question of lung clearance is of great scientific importance. Its onset takes place during gestation (when?) and continues immediately p.n. However, this process is not directly related to the CT studies performed and described here. Indeed, the authors support the reviewer's request to investigate how lung clearance in pups obtained by caesarean section differs from that seen in pups born per vias naturales and having taken the passage through the apertura pelvis; in this Ms. this question is unresolved, not least because of the extremely small number of cases (n=1 CC), but should be urgently addressed in follow-up studies.

Some of the major comments and criticisms include:

/

Title:

/

4.      

I understand the meaning of “histomorphometric” analysis of the lung. However, morphometry encompasses a quantitative study of lung histologic architecture. Authors have not performed a quantitative analysis, but a descriptive histological evaluation of the lung components. Please, revise the title terminology

lines 15-17 and 32-34 “histomorphometric” was replaced by “histopathologic”, “histopathologic tissue classification techniques”

Simple Summary

5.      

Line 15: please clarify the meaning of “ventilation”. One can interpret as artificial lung insufflation other than natural pulmonary expansion.

line 15: “ventilation” was replaced by “aeration” and in lines 17-18 by “aerated” neonatal lung tissue; ventilation of the implies typically the homogeneous active ventilation of the lung – reanimation is only pressure and thorax compression and no artificial insufflation in contrast to the terminology and practice in human medicine.

Abstract

6.      

Line 37-38: Your conclusion is not paired with your objectives. Please, reformulate the conclusion in order to answer the scientific question established at the objective. In addition, conclusion is too overstated based on what authors have really performed in their methodology.

“The results of this study are in particular clinically relevant, and suggest that resuscitation measures should be performed consistently and intensively within the first 10 min p.n., preferably with the pup in the abdomen-chest position, to achieve the highest possible level of venti-lated pulmonary tissue.”

We apologize, but we do not understand this comment - please specify and refer to a specific sentence or statement in the Conclusions. The 10 min p.n. result directly from the determined HU measured values. Accordingly, the early resuscitation of low-life pups is justified. The appropriate body position results from the differences dorsal/ventral ventilation.

Introduction:

7.      

Although the Introduction is well written, authors should try to address thoroughly the respiratory physiology of the neonatal period.

We have cited Mortola [4] as key literature and cannot review the entire literature on neonatal pulmonary physiology due to space limitations.

8.      

Line 57 and 66-68: these statements need a reference for canine neonates

The references [1,2] and [4] were cited in this context.

9.      

Lines 60-61: New data from Regazzi et al Morphometric and functional pulmonary changes of premature neonatal puppies after antenatal corticoid therapy (2020) show that lung alveolarization starts at the final stages of fetal development. Thus, alveoli can be found in recent born puppies.

This reference was  now included in the discussion:

In dogs, cranial pulmonary lobes of the fetus are more developed than caudal lobes, as well as in a centrifugal manner [20]. Lung alveolarization starts at the final stages of fetal development [21].

10.   

Lines 70-71: Again, results from Regazzi et al. (2020) show that surfactant protein SP-B can be detected in lung parenchyma early as 55 days of gestation.

This specific information can not be found in the cited paper for the (non-treated) control group. Please specify.

The work of Regazzi et al does not effectively indicates, from when how much surfactant is formed in canine fetuses.  Here it is shown that in the preterm - control - group (n = 5, gravidity interrupted 55 - 57 day p.cohabitationem) a low percentage of pneumocytes II is present. This is clearer in the term - group (p < o.o5 ). This would perhaps be suitable for citation (see also resp. to #36)

11.   

Line 75: above the lung functioning during the neonatal transition period, pulmonary clearance is of utmost importance to achieve full pulmonary gas exchange. 

This reference does not play a role in the targeted investigations; however, this is truly important for comparison between CC and naturally born pups. All in all, even published data are very vague and indeterminate, exact data are not given in Regazzi et al (2020). Overall, the point is that fluid and foreign matter are cleared from the upper airways by the cilia - The fluid is essentially completed in normal born pups by compression of the thorax as it slides through the apertura pelvis, continued as it passes through the narrow vaginal canal . When the thorax of the fetus has overcome the last inhibition threshold (vulva), it spontaneously dilates - this is then the first act of inspiration. So the puppy starts breathing even before the birth is completed. The final resulting problem is the posterior end position and the cesarean section. But this is in no matter part of this manuscript, also not in the discussion.

12.   

Line 78: how exactly did authors perform the evaluation of gas exchange capacity? By reading the methodology, the primary postnatal function of the respiratory system to provide sufficient oxygenation and to eliminate CO2 was not analyzed in the present manuscript. Please, clarify

As part of the clinical examination of the newborns, the pH, BE and HU were correlated for time points 0,6,12,24 h - unfortunately, however, these data are not available for all 3 litters as well as for all pups. Therefore, the authors have decided not to reproduce them in this manuscript because of their incompleteness.

13.   

Line 79: see comment above for the term “ventilation”

line 80: although we cannot follow the reviewer’s opinion without reservation, the term has been replaced by “aeration”

Material and methods:

Some very important information should be provided:

14.   

-        How exactly the gestation period was calculated? Based on ovulation or LH peak?

Gestation period was calculated by using vaginal cytology and P4 test before mating, from day 1 of mating the period was calculated – in Germany in principle LH peak determination is unusual and was not applied due to its inherent uncertainties.

15.   

-        Were all births vaginal deliveries?

See Table 1: only 1/40 pups was delivered by Caesarean section

16.   

-        What was the cause of death of stillbirth puppies? Was a necropsy performed?

Causes of death were not determined by autopsy in the stillborn pups. Exitus letalis intra partum,
See Question #6

17.   

-        The statistical analysis is too shallow. Please, include the probability values used to consider significance (P).

we do not agree that the statistical analysis is too shallow. Line 180: “As significance level p<0.05 was used.” was added to clarify

18.   

Line 98: how the respiratory distress syndrome was diagnosed?

Typical of NRDS is dyspnea associated with cyanotic mucous membranes and a still present heart or pulse. Apart from the typical signs of NRDS, the modified Apgar score can be used to classify the severity of NRDS in canine and feline puppies more concisely in a very short time, providing more specific indications for the emergency measures to be taken. Additional laboratory tests were performed to determine blood pH (> 7.2; 7.1-7.0; < 7.0) and current base excess (BE) (0 to +5 mmol/l; ± 0 to -10; < -10 mmol/l).

19.   

Line 99: please detail the medical procedures executed

All bitches were under continuous veterinary care during gravidity. On the 50th day of gestation an X-ray was taken to determine the number of fetuses (so that preparations could be made). From the 58th day the temperature of the body core was measured twice a day, the degree of edema of the vulva was determined, the milk infusion in the mammary complexes was checked by palpation, all in order to determine the time of the beginning of the birth. The birth took place in an extra room with a whelping box (temperature in the first 7 days after birth 29 degrees, after that 24 degrees in the puppy compartment). No veterinary intervention was necessary during the birth (= normal birth). Neonatal vitality was determined immediately p.n. by APGAR score, lungs were auscultated on all four quadrants, HF was measured. This was followed by a segmental examination (exclusion of externally visible malformations).

APGAR score, BE,

20.   

Line 104: the inclusion of a surgical born puppy biases the work, as recently shown by Almeida et al. Both spontaneous vaginal delivery and elective caesarean section influence neonatal redox status in dogs (2022)

The authors are aware of the problem described. However, despite very low case numbers, another postnatally ventilated lung should be integrated into the study for the histopathological examinations and evaluations. For the aforementioned ethical reasons, only pups that had died of natural causes were available for this purpose.

21.   

Line 110: I am not sure that using the term “control” is the correct nomenclature for this situation. Perhaps it will be better to name it as “non-insufflated control” or “negative control”

The term control was deleted and the paragraph was rephrased: “In addition, 6 other stillborn pups (total number of stillbirths: 8) and one other vital depressed neonate (exitus 30 min p.n.) from other litters of dams of the same body weight class (> 20 kg) were included in the studies for comparison purposes for morphological imaging of lung tissue.”

22.   

Lines 117-118: I do not think that ethical approve has to be solely for experimental animals. The consent of an ethical committee has to be obtained regardless

See #4: no experimental approach with need for ethical committee – everything took place with the consent of all owners o.)

23.   

Results:
Line 188: Perhaps it would be illustrative to include the CT comparative images.

We appreciate very much this comment and included CT images for stillborn and viable pups in the now new Figure 1

24.   

Line 212: Figure 4C

Line 209-210: Figure 6

Discussion

25.   

Line 237-238: if a functional analysis of blood oxygenation was not performed, you cannot affirm that gas exchange is being fully achieved.

See 53. As part of the clinical examination of the newborns, the pH, BE and HU were correlated for time points 0,6,12,24 h - unfortunately, however, these data are not available for all 3 litters as well as for all pups. Therefore, the authors have decided not to reproduce them in this manuscript because of their incompleteness.

26.   

Line 240-241: the use of stillborn as control animals is a matter of doubt. Is it possible to affirm that fetal death occurred during whelping? Can you exclude premature fetal death? Thus, the atelectasic aspect of the lung may be only a consequence of immature lung development in a canalicular phase (please see Sipriani et al., Pulmonary maturation in canine foetuses from early pregnancy to parturition 2009)

Because the stillborn puppies were not subjected to further pathological-anatomical examination, nothing can be said about the time of death and its cause. Only the lungs were submitted to a histopathological examination.

27.   

Lines 255-257: this statement needs reference

Lines 280-282? Which lines are meant? The original line numbers probably got due to formatting issues in disarray. Please refer to specific sentences.

28.   

Line 259: saccular

Line 284 – was replaced

29.   

Line 272-274: According to the study of Sipriani et al. (2009), cranial pulmonary lobes of the canine fetus are more developed than caudal lobes, as well as in a centrifugal manner.

Lines 297-299 This sentence and reference was included “In dogs, cranial pulmonary lobes of the fetus are more developed than caudal lobes, as well as in a centrifugal manner [20].”

30.   

Lines 279-281: please refer to the manuscripts of Sipriani et al. (2009) and Regazzi et al. (2020).

Lines 279-281? Which lines are meant? They probably got due to formatting issues in disarray. Please refer to specific sentences.

31.   

Lines 289-291: this statement needs reference

dito

32.   

Line 298: perhaps it would better to say "inflate or insufflate" instead of “gas exchange”

Actually we disagree since “gas exchange” is the usually used terminology in this context

Conclusion

33.   

The conclusion should be mirrored by the objectives. Although very useful, future application of the data should be stated, but it cannot be considered the conclusion of the research. Please, rewrite.

The results of this study have clinical relevance in that, particularly in the context of necessary resuscitation measures, these should be performed consistently and intensively within the first 10 min p.n., preferably with the pup in the abdomen-chest position, in order to achieve the highest possible degree of ventilated pulmonary tissue.

Round 2

Reviewer 1 Report

Dear Authors,

Thank you for reviewing the manuscript and addressing my concerns. Nonetheless, in my opinion, the paper needs still improvement:

1.

In their manuscript "Computed tomographic and histomorphometric studies of lung function immediately post natum in canine neonates" the Authors try to describe the development of lungs in dogs in short period after birth.

/

2.

Although the topic is important and can influence the safety of canine neonates, in my opinion the study has serious flaws that do not allow a publication in the presented form.

/

3.

In the aims of their study Authors mention the description of histopathological image of the lungs in dogs dying during the first 24h of life (aim number 2); nonetheless, the number of dogs examined histologically and compared to stillborn pups is very small (n=3) and therefore in my opinion do not allow to draw any significant conclusions, as the possibility of error is relatively high. I understand that obtaining a higher number of dogs dying shortly after birth can be challenging, but it do not dispense us from publishing valuable research and results. Therefore, either the histological and morphometric study should not be mentioned as an aim of the study (and in the title of the paper), or the number of examined animals should be increased.

- Title: “histomorphometric” was replaced by “histopathologic”

- Aim: #2 was deleted and histopathology as method integrated in the aim (#1): “...to demonstrate the development of pulmonary tissue ventilation aeration within the first period of adaptation in vital born canine neonates by CT, and using also histopathological methods.

Since histopathology is an integral and important part of this study we do not agree to skip it completely from the title, abstract or aim.

I do not agree with your statement that the histological examination should be mentioned in the title of the paper; in my opinion the number of examined animals do not allow to draw any significant conclusions; moreover, if you state the histological examination as a aim of your study, in my opinion, this aim should be considered as not fulfilled due to the small number

4.

Secondly, in lines 117-118 Authors mention that the CT examination was a part of extended examination procedure and therefore did not require the approval of ethics committee and that the imaging examination helped to ensure the survival of the pups. I do not understand how the examination could help in the survival of animals? Is the double CT examination (directly after birth and after 24 h) used in the clinic, where the study was performed, as a routine practice in newborn dogs? If not, in my opinion, at least a statement from the local ethical committee should have been obtained to avoid misdoubts. If the Authors' standpoint is based on the state law (as it can differ between the countries), please indicate that in the text with appropriate reference.

Under file number 7221.3-2.039/13, the competent authority (LALLF Mecklenburg-Vorpommern, Rostock) certified that the project did not require approval under Section 8a (1) (2) of the Animal Protection Act (in 2013 when the first experiments started).

This was added lines 130 ff.

Thank you for that explanation

5.

Next, the study was conducted using two various CT devices. The Authors do not clarify if there were significant differences in the two devices that could influence the results. This should be either clarified or mentioned in the limitation section.

line 138: This was clarified by “However, there were no technical differences in the two devices that could influence the results.”

6.

Moreover, the causes of animals' death prior or after birth are not recognised and therefore should be mentioned as a study limitation, as various factors (mainly infectious agents) can influence the development of the respiratory system during pregnancy.

It was not the aim of the study to determine the causes of death in stillbirths - all stillbirths were so-called newborn pups that died during birth. This may be a consequence of birth length, overlong inter-pup interval. The specific causes could not be determined clinically. There were no indications of infectious causes or malformations.

I also do not agree with the statement that there were no indications of infectious causes or malformations - stillbirth is an indication of a problem - it can be, as you mentioned a consequence of birth length, overlong inter-pup interval, but also a consequence of an infection of a bitch of the puppies; therefore if you did not perform appropriate tests, you should mention it in the limitation section

7.

Hereafter I list also the minor revisions that should be addressed:

/

8.

- line 33: it is not clear what the abbreviation "p.n" states for in this sentence? Did the Authors mean "3 early post natum neonates"?

Was rephrased: “(8 stillborn and 3 neonates, died early post natum, respectively)”

9.

- throughout the text: please change reference format from [1], [2] to [1,2], etc.

Was done accordingly

10.

- lines 68-69: This sentence is not clear - please rephrase

Was rephrased to: “Bitches in which the number of fetuses exceeds the normal breed average by more than 20% will give birth earlier than those with a below average litter size.”

11.

- lines 96-100: was the CT examination in those two pups performed before or after the resuscitation? Or was it not performed at all? It is not clear.

line 107: rephrased “This necessary resuscitation consisted of physiotherapeutic and medicinal procedures and was performed before CT examination [5].”

12.

- lines 126 and further: what was the duration of the CT examination?

CT took 80-90 sec (at max. 120 sec.) – was added in lines150-151

13.

- lines 126-136: the group design is not clearly presented; I believe that the animals (n=31) were divided into three groups and then re-examined after 24 hours but it is not described properly; basing on the information provided by the Authors the study was conducted on 62 animals (G1 n=11, G2 n=15, G3 n=5, G4 n=31)

This was rephrased and “group 4” more clearly described as: “All these animals were re-examined by CT at 24 h p.n.”

14.

- line 170: in my opinion the values obtained in the stillborn pups (reference) should be also presented in the Figure 1

The results shown in Figures 2 and 3 refer to the repeated measures ANOVA (thus estimates), in which only the results of the live-born animals were included, because in the stillborn animals the ventilation of the lungs could not be recorded at different time points. (this explanation was also added in lines 201ff)

15.

- lines 169-194: please consider combining section 3.1 and 3.2 as the results obtained in stillborn pups are used as a reference for alive ones and are mentioned in the section 3.1

Was done as recommended

16.

- lines 177-178: it is not clear in which time point were this results taken: does it refer to measurements obtained in ≤10 min, 10-15 min and 16-20 min for group 1, 2 and 3, respectively (if yes, it should be clearly stated and the results should be noted for each of the groups) or to the measurements obtained 24h after birth?

Thank you for clarification in the manuscript

17.

- Figure 1: are the measurements taken 24h after birth presented in the figure? It is not clear for me, as the figure present group 1, group 2 and group 3

See #16.

Thank you for clarification in the manuscript

18.

- Figure 2: as mentioned previously - what was the time of those measurements?

It was clarified by: “directly post natum (time p.n. groups 1,2,3; n=31)

The results shown in Figure 3 refer also to the repeated measures ANOVA (thus estimates), in which only the results of the live-born animals were included, because in the stillborn animals the ventilation of the lungs could not be recorded at different time points.

Thank you for clarification in the manuscript

19.

- lines 195-212: as previously, please consider combining section 3.3 and 3.4 as one

Was done as recommended

20.

- line 208: please explain the abbreviation "NANS"

was replaced by “NRDS”

21.

- lines 211-212: this description reflects the histopathological image observed in stillborn pups and presented in Figure 5 - please clarify

the description actually refers to figure 6; “3C” remained from an earlier version of the manuscript and is incorrect, was replaced

22.

- line 212, parenthesis: please correct to Figure 4C

3C was replaced by 6

23.

- Figure 3: please add a reference to Fig. 3 in the text; please clarify if the Fig. 3 present the same data as Fig. 2 but in each studied group? Was there a difference in the results observed between the groups?

These are the “measured HU” and not the estimated values and therefore not identic to figure 3

“The measured HU are depicted in Figure 4.” Was added in line 204

24.

- Figure 4, legend: in the description of C please rephrase to "C) Ventral lobe, death 48h p.n...." to keep concise with the rest of figure legend

Was done accordingly

25.

- Figure 6: there is no reference to Fig 6 in the manuscript - please add

Ref. was added in lines 217-218: However, some areas of the pulmonary tissue showed moderately aerated lung saccules and dilated bronchioles (Figure 6).

26.

- Figure 7: there is no reference to Fig. 7A in the manuscript - please add; Figure 7B do not correspond with the legend of Figure 7 - please either rephrase the legend of Fig. 7 to include the difference in dorsal vs. ventral lobes in stillborn pups or divide Fig. 7 to two separate figures.

Reference Figure 8A, 8B was added in line 221-224

The legend for 8B. is correct as it stands and corresponds to the figure for the stillborn pups: “There are no anatomically based differences in stillborn pups with respect to optically empty area in lung tissue of dorsal and ventral regions (P = 0.2353).”

27.

- lines 282-292: I appreciate that the mentioned facts "are known" to the Authors, but still some references confirming those statements would be useful; otherwise it is just a hypothesis and not a discussion with the literature

Potentially the line numbering has changed due to format issues? This comment is not specific enough since as it stands, since in lines 282-292 we cite [9,15,16,17] which we consider sufficient. If we missed the point of the reviewer here, please specify the “facts” and check line numbers again.

I was referring to the paragraph: "It is known that the first breaths of the canine neonate begin before completion of expulsion to compensate for the no longer intact materno-placental gas exchange sub partu. The thoracic compression caused by the narrowness of the soft birthway is released when the head and thoracic segments have passed the rima vulvae and the first breaths occur, leading to expansion of the thorax, inflation of the pulmonary alveoli, and hence- forth regular inspiration and expiration. As a result of the inflation, the cubic epithelial cell association of the bronchiolar ducts is ruptured into metameric cell clusters between which alveolar sacs with alveoli are immediately formed. As a special feature of canine neonates, their lungs can remain in an apneic inspiratory position for up to 20 seconds after the inflow of the first air stream. This apparently causes more distant alveoli in the terminal tip lobes of the lung to dilate, allowing more intensive gas exchange before expiration begins. “ (now in lines 301-316) - please add references

28.

- lines 294-296: as above - please add references to that statement

Again, perhaps wrong line numbers? For 294-296 we cited [8]. Or specify, which statement does the reviewer mean in detail.

Lines 317-319 after revision:

“Yet another circumstance should be noted regarding the development of the first pul- 317 monary gas exchange in canine neonates. Due to the variable gestation period, it can be 318 assumed that the lung tissue has a different developmental state immediately post natum.”

29.

- lines 298-299: Group 3 was examined up to 20 minutes after birth; what is the basis of the statement that 82% of the lungs is inflated up to 30 minutes p.n.?

Was changed to 20 min; Howevr, 30 min result from long clinical experiences for reanimation of neonatal canines

30.

- line 299: the parenthesis is unnecessary

The line numbers seem to be in disarray, please specify which parenthesis is meant

Lines 321-322:

“Up to 20 min p.n., ventilation increases 321 further, so that 82% of the pulmonary area is already ventilated (Figure 2).”

You already mention the results in the results section - in my opinion it is not necessary to mention it also in the discussion section

31.

- lines 300-301: please rephrase this sentence

If lines 320-321 are meant? “It is these findings that would need to be considered when resuscitating canine neonates suffering from neonatal respiratory depression” This was rephrased to “These findings need to be considered when resuscitating neonates with neonatal respiratory depression.”

32.

- lines 300-306: the conclusion drown in this paragraph is not confirmed by the conducted study; the Authors did not perform studies on the impact of the time of resuscitation on the lungs inflation, they only stated that the lungs inflate in approx. 73% during the first 10 minutes after birth (which can be identified with the first breath). But the Authors only examined the inflation in normal lungs in healthy puppies. The respiratory depression can influence the lung capacity and duration of full inflation, nonetheless also a situation is possible when the full inflation will be obtained even if the first breath is forced few minutes after birth. It should be examined in controlled manner and only then such conclusions can be drawn

Are lines 318 meant?

The studies have shown that the first 10 min are the most important for lung development. Our investigations carried out have provided the physiological basis for studying the most appropriate resuscitation measures in further studies. Further investigation is undoubtedly important but not the focus of this study.

Lines 323-329: “These findings need to be considered when resuscitating neonates with neonatal respiratory depression.. Thus, it is important that appropriate targeted physical resuscitation procedures are applied in the first 10 min p.n. to ventilate as much of the lung tissue as quickly as possible. Because of the differences in the degree of ventilation described here, resuscitation with the pup in the thoracic position seems more appropriate. These measures should then be continued for up to 30 min p.n., depending on the condition. 

The onset point and duration of the resuscitation was not the aim of the study, therefore the conclusions are not supported by the results

33.

- line 306: as above: what is the basis of limitation of 30 minutes of resuscitation; the Authors did not examine the impact of starting point and duration of resuscitation on the pulmonary capacity and inflation

this was not the aim of the work to optimize the resuscitation methods - completely different question

Lines 328-329 “These measures should then be continued for up to 30 min p.n., depending on 328 the condition. “

I agree that it was not the aim of your study therefore why do you draw that conclusion?

34.

- line 312-313: this statement is not supported by the results of the study (only one puppy was delivered by CC in the study)

lines 327-328? Was rephrased to: “It can be assumed that this recommendation is particularly relevant in neonates who developed through obstetric laparotomy.”

The situation of lung development in CC pups would have to be controlled in another study, which does not get its lung fluid squeezed out per vias naturales. This would also be a very extensive further study which is not the aim of this work. - it is a crucial hint that is gratefully received but cannot be implemented in this manuscript.

I was reffering to “it would be necessary to consider whether the neonate was born per vias 335 naturalis or by obstetric laparotomy.” (lines 335-336 now)

Again, the conclusion is not supported by the results

35.

- lines 313-318: I find it difficult to follow whether this paragraph is related to the impact of the number of foetuses on the duration of pregnancy or on the way of birth (natural or CC)

line 358: it is highly significant and results from literature – has nothing to do with way of birth only length of gestation – questionable if longer gestation results in better development of the lungs

"It has also been demonstrated that in this species, gestation length depends on the litter size [6,7]. Bitches in which the number of fetuses exceeds the normal breed average are more likely to give birth than those with a below- average number of fetuses. In boxer bitches, gestation length therefore varies considerably (55 to 71 days). Conversely, it can be assumed that the lung tissue also shows a different developmental state depending on the gestation period.” (lines 336-341 now)

Did the Authors mean that in pregancies smaller than average, the duration is prolonged, or CC is required?

36.

- lines 325 (345?): what is the relation of the presence of surfactant in the lambs' lungs to the rest of the paragraph? I find it unclear

In canine pups, surfactant protein SP-B can be detected in the lung parenchyma as early as 55 days of gestation [21].

37.

- lines 326-342: please add references to parts of this paragraph (and not only at the end)

Which lines are meant? Please refer to specific sentences

In 326-342 the refs [6,7] were cited which cover the complete paragraph

“At birth, some of the fetal lung fluid is forced out of the bronchial tree during thoraciccompression as the fetus passes through the maternal pelvis and vagina and is either apparently swallowed by the neonate or becomes lodged in and obstructs the upper airway“ and further on

38.

- lines 343-347: according to previously mentioned doubts the conclusions are unfounded and exaggerated; basing on their study, the Authors cannot draw conclusions on the efficacy of resuscitation depending on the time of its initiation and duration

Which lines are meant? Please refer to specific sentences

I meant the conclusions. They are not supported by the methodology or results:

“The results of this study have clinical relevance in that, particularly in the context of necessary resuscitation measures, these should be performed consistently and intensively within the first 10 min p.n., preferably with the pup in the abdomen-chest position, in order to achieve the highest possible degree of ventilated pulmonary tissue. The investigations carried out here have provided the basis for evaluating the most appropriate resuscitation measures in further studies.”

39.

- References: almost half of the cited references (11 out of 23) are older than 1990 and three further are from 1990's. In my opinion newer references should be discussed to show the importance of the research to 21st century veterinary science

We have now cited Regazzi et al. and Sipriani et al. as recent papers; however, even these unfortunately only cite older literature because not much new exists

Sipriani, T. M.; Grandi, F.; da Silva, L. C.; Maiorka, P. C.; Vannucchi, C. I. Pulmonary maturation in canine foetuses from early pregnancy to parturition. Reprod Domest Anim. 2009, 44 Suppl 2, 137–140.

Regazzi, F. M.; Silva, L. C. G.; Lúcio, C. F.; Veiga, G. A. L.; Angrimani, D. S. R.; Vannucchi, C. I. Morphometric and func-tional pulmonary changes of premature neonatal puppies after antenatal corticoid therapy. Theriogenology. 2020, 153, 19–26.

Thank you for that clarification.

Author Response

The authors would like to thank reviewer #1 for his careful study and numerous and helpful suggestions for improvement, which we have implemented to the best of our ability.

Reviewer #1

Rebuttal 1st round

Rebuttal 2nd round

1.

In their manuscript "Computed tomographic and histomorphometric studies of lung function immediately post natum in canine neonates" the Authors try to describe the development of lungs in dogs in short period after birth.

/

/

2.

Although the topic is important and can influence the safety of canine neonates, in my opinion the study has serious flaws that do not allow a publication in the presented form.

/

/

3.

In the aims of their study Authors mention the description of histopathological image of the lungs in dogs dying during the first 24h of life (aim number 2); nonetheless, the number of dogs examined histologically and compared to stillborn pups is very small (n=3) and therefore in my opinion do not allow to draw any significant conclusions, as the possibility of error is relatively high. I understand that obtaining a higher number of dogs dying shortly after birth can be challenging, but it do not dispense us from publishing valuable research and results. Therefore, either the histological and morphometric study should not be mentioned as an aim of the study (and in the title of the paper), or the number of examined animals should be increased.

- Title: “histomorphometric” was replaced by “histopathologic”

- Aim: #2 was deleted and histopathology as method integrated in the aim (#1): “...to demonstrate the development of pulmonary tissue ventilation aeration within the first period of adaptation in vital born canine neonates by CT, and using also histopathological methods.

Since histopathology is an integral and important part of this study we do not agree to skip it completely from the title, abstract or aim.

I do not agree with your statement that the histological examination should be mentioned in the title of the paper; in my opinion the number of examined animals do not allow to draw any significant conclusions; moreover, if you state the histological examination as a aim of your study, in my opinion, this aim should be considered as not fulfilled due to the small number

The title was now changed accordingly to: Computed tomographic studies of lung function immediately post natum in canine neonates

Comment: We do not support the reviewr’s opinion since the number of investigated animals is not only crucial for the valid of a sound histopathological investigation. The correlation of CT and morphology by light microscopy is a unique feature of this manuscript. However, to comply with the reviewer’s individual desire we fulfilled his request.

4.

Secondly, in lines 117-118 Authors mention that the CT examination was a part of extended examination procedure and therefore did not require the approval of ethics committee and that the imaging examination helped to ensure the survival of the pups. I do not understand how the examination could help in the survival of animals? Is the double CT examination (directly after birth and after 24 h) used in the clinic, where the study was performed, as a routine practice in newborn dogs? If not, in my opinion, at least a statement from the local ethical committee should have been obtained to avoid misdoubts. If the Authors' standpoint is based on the state law (as it can differ between the countries), please indicate that in the text with appropriate reference.

Under file number 7221.3-2.039/13, the competent authority (LALLF Mecklenburg-Vorpommern, Rostock) certified that the project did not require approval under Section 8a (1) (2) of the Animal Protection Act (in 2013 when the first experiments started).

This was added lines 130 ff.

Thank you for that explanation

In addition to this certificat an updated statement from the competent authority was  received on request 14.04.2023 with file number 7221.3-17493_18059_23 clearly stating that

”After appropriate examination, I can inform you that in our view this is not an animal experiment in the sense of Art. 7 para. 2 Animal Welfare Act.”

Therefore, according the German legislation an ethical committee is not involved. This was now also updated in lines 131ff.

We added the following paragraph in lines 490 ff:  Statement on animal protection: In addition to the citetd statement of the Competent Authority, we declare that the examination of a procedure that is not an animal experiment in the sense of the German Animal Welfare Act by an ethics committee is not legally required in Germany by the Animal Welfare Act or other regulations.

5.

Next, the study was conducted using two various CT devices. The Authors do not clarify if there were significant differences in the two devices that could influence the results. This should be either clarified or mentioned in the limitation section.

line 138: This was clarified by “However, there were no technical differences in the two devices that could influence the results.”

/

6.

Moreover, the causes of animals' death prior or after birth are not recognised and therefore should be mentioned as a study limitation, as various factors (mainly infectious agents) can influence the development of the respiratory system during pregnancy.

It was not the aim of the study to determine the causes of death in stillbirths - all stillbirths were so-called newborn pups that died during birth. This may be a consequence of birth length, overlong inter-pup interval. The specific causes could not be determined clinically. There were no indications of infectious causes or malformations.

I also do not agree with the statement that there were no indications of infectious causes or malformations - stillbirth is an indication of a problem - it can be, as you mentioned a consequence of birth length, overlong inter-pup interval, but also a consequence of an infection of a bitch of the puppies; therefore if you did not perform appropriate tests, you should mention it in the limitation section

The following sentence were now included in lines 117f:

Because no additional microbiological studies were performed on the pathogenesis of stillbirth, this limits interpretation to some extent.

And in lines 302ff:

For comparison, stillborn pups were included in the first measurement period. The cause of stillbirth was not investigated and limits therefore interpretation as reference. It could be consequence of birth length, overlong inter-pup interval, but also a sequela of an infection of the dam.

Reference [14] was inserted here too.

7.

Hereafter I list also the minor revisions that should be addressed:

/

/

8.

- line 33: it is not clear what the abbreviation "p.n" states for in this sentence? Did the Authors mean "3 early post natum neonates"?

Was rephrased: “(8 stillborn and 3 neonates, died early post natum, respectively)”

/

9.

- throughout the text: please change reference format from [1], [2] to [1,2], etc.

Was done accordingly

/

10.

- lines 68-69: This sentence is not clear - please rephrase

Was rephrased to: “Bitches in which the number of fetuses exceeds the normal breed average by more than 20% will give birth earlier than those with a below average litter size.”

/

11.

- lines 96-100: was the CT examination in those two pups performed before or after the resuscitation? Or was it not performed at all? It is not clear.

line 107: rephrased “This necessary resuscitation consisted of physiotherapeutic and medicinal procedures and was performed before CT examination [5].”

/

12.

- lines 126 and further: what was the duration of the CT examination?

CT took 80-90 sec (at max. 120 sec.) – was added in lines150-151

/

13.

- lines 126-136: the group design is not clearly presented; I believe that the animals (n=31) were divided into three groups and then re-examined after 24 hours but it is not described properly; basing on the information provided by the Authors the study was conducted on 62 animals (G1 n=11, G2 n=15, G3 n=5, G4 n=31)

This was rephrased and “group 4” more clearly described as: “All these animals were re-examined by CT at 24 h p.n.”

/

14.

- line 170: in my opinion the values obtained in the stillborn pups (reference) should be also presented in the Figure 1

The results shown in Figures 2 and 3 refer to the repeated measures ANOVA (thus estimates), in which only the results of the live-born animals were included, because in the stillborn animals the ventilation of the lungs could not be recorded at different time points. (this explanation was also added in lines 201ff)

/

15.

- lines 169-194: please consider combining section 3.1 and 3.2 as the results obtained in stillborn pups are used as a reference for alive ones and are mentioned in the section 3.1

Was done as recommended

/

16.

- lines 177-178: it is not clear in which time point were this results taken: does it refer to measurements obtained in ≤10 min, 10-15 min and 16-20 min for group 1, 2 and 3, respectively (if yes, it should be clearly stated and the results should be noted for each of the groups) or to the measurements obtained 24h after birth?

We tried to clarify by changing the Figure 2 legend

Thank you for clarification in the manuscript

/

17.

- Figure 1: are the measurements taken 24h after birth presented in the figure? It is not clear for me, as the figure present group 1, group 2 and group 3

See #16.

Thank you for clarification in the manuscript

/

18.

- Figure 2: as mentioned previously - what was the time of those measurements?

It was clarified by: “directly post natum (time p.n. groups 1,2,3; n=31)

The results shown in Figure 3 refer also to the repeated measures ANOVA (thus estimates), in which only the results of the live-born animals were included, because in the stillborn animals the ventilation of the lungs could not be recorded at different time points.

Thank you for clarification in the manuscript

/

19.

- lines 195-212: as previously, please consider combining section 3.3 and 3.4 as one

Was done as recommended

/

20.

- line 208: please explain the abbreviation "NANS"

was replaced by “NRDS”

/

21.

- lines 211-212: this description reflects the histopathological image observed in stillborn pups and presented in Figure 5 - please clarify

the description actually refers to figure 6; “3C” remained from an earlier version of the manuscript and is incorrect, was replaced

/

22.

- line 212, parenthesis: please correct to Figure 4C

3C was replaced by 6

/

23.

- Figure 3: please add a reference to Fig. 3 in the text; please clarify if the Fig. 3 present the same data as Fig. 2 but in each studied group? Was there a difference in the results observed between the groups?

These are the “measured HU” and not the estimated values and therefore not identic to figure 3

“The measured HU are depicted in Figure 4.” Was added in line 204

/

24.

- Figure 4, legend: in the description of C please rephrase to "C) Ventral lobe, death 48h p.n...." to keep concise with the rest of figure legend

Was done accordingly

/

25.

- Figure 6: there is no reference to Fig 6 in the manuscript - please add

Ref. was added in lines 217-218: However, some areas of the pulmonary tissue showed moderately aerated lung saccules and dilated bronchioles (Figure 6).

/

26.

- Figure 7: there is no reference to Fig. 7A in the manuscript - please add; Figure 7B do not correspond with the legend of Figure 7 - please either rephrase the legend of Fig. 7 to include the difference in dorsal vs. ventral lobes in stillborn pups or divide Fig. 7 to two separate figures.

Reference Figure 8A, 8B was added in line 221-224

The legend for 8B. is correct as it stands and corresponds to the figure for the stillborn pups: “There are no anatomically based differences in stillborn pups with respect to optically empty area in lung tissue of dorsal and ventral regions (P = 0.2353).”

/

27.

- lines 282-292: I appreciate that the mentioned facts "are known" to the Authors, but still some references confirming those statements would be useful; otherwise it is just a hypothesis and not a discussion with the literature

Potentially the line numbering has changed due to format issues? This comment is not specific enough since as it stands, since in lines 282-292 we cite [9,15,16,17] which we consider sufficient. If we missed the point of the reviewer here, please specify the “facts” and check line numbers again.

I was referring to the paragraph: "It is known that the first breaths of the canine neonate begin before completion of expulsion to compensate for the no longer intact materno-placental gas exchange sub partu. The thoracic compression caused by the narrowness of the soft birthway is released when the head and thoracic segments have passed the rima vulvae and the first breaths occur, leading to expansion of the thorax, inflation of the pulmonary alveoli, and hence- forth regular inspiration and expiration. As a result of the inflation, the cubic epithelial cell association of the bronchiolar ducts is ruptured into metameric cell clusters between which alveolar sacs with alveoli are immediately formed. As a special feature of canine neonates, their lungs can remain in an apneic inspiratory position for up to 20 seconds after the inflow of the first air stream. This apparently causes more distant alveoli in the terminal tip lobes of the lung to dilate, allowing more intensive gas exchange before expiration begins. “ (now in lines 301-316) - please add references

The following references were now included: [2,5]

28.

- lines 294-296: as above - please add references to that statement

Again, perhaps wrong line numbers? For 294-296 we cited [8]. Or specify, which statement does the reviewer mean in detail.

Lines 317-319 after revision:

“Yet another circumstance should be noted regarding the development of the first pul- 317 monary gas exchange in canine neonates. Due to the variable gestation period, it can be 318 assumed that the lung tissue has a different developmental state immediately post natum.”

This sentence was rephrased since this is the author’s opinion: Due to the variable gestation period, it can be assumed hypothesized that the lung tissue has a different developmental state immediately post natum.

29.

- lines 298-299: Group 3 was examined up to 20 minutes after birth; what is the basis of the statement that 82% of the lungs is inflated up to 30 minutes p.n.?

Was changed to 20 min; Howevr, 30 min result from long clinical experiences for reanimation of neonatal canines

/

30.

- line 299: the parenthesis is unnecessary

The line numbers seem to be in disarray, please specify which parenthesis is meant

Lines 321-322:

“Up to 20 min p.n., ventilation increases 321 further, so that 82% of the pulmonary area is already ventilated (Figure 2).”

You already mention the results in the results section - in my opinion it is not necessary to mention it also in the discussion section

The parenthesis was deleted as recommended.

31.

- lines 300-301: please rephrase this sentence

If lines 320-321 are meant? “It is these findings that would need to be considered when resuscitating canine neonates suffering from neonatal respiratory depression” This was rephrased to “These findings need to be considered when resuscitating neonates with neonatal respiratory depression.”

/

32.

- lines 300-306: the conclusion drown in this paragraph is not confirmed by the conducted study; the Authors did not perform studies on the impact of the time of resuscitation on the lungs inflation, they only stated that the lungs inflate in approx. 73% during the first 10 minutes after birth (which can be identified with the first breath). But the Authors only examined the inflation in normal lungs in healthy puppies. The respiratory depression can influence the lung capacity and duration of full inflation, nonetheless also a situation is possible when the full inflation will be obtained even if the first breath is forced few minutes after birth. It should be examined in controlled manner and only then such conclusions can be drawn

Are lines 318 meant?

The studies have shown that the first 10 min are the most important for lung development. Our investigations carried out have provided the physiological basis for studying the most appropriate resuscitation measures in further studies. Further investigation is undoubtedly important but not the focus of this study.

Lines 323-329: “These findings need to be considered when resuscitating neonates with neonatal respiratory depression.. Thus, it is important that appropriate targeted physical resuscitation procedures are applied in the first 10 min p.n. to ventilate as much of the lung tissue as quickly as possible. Because of the differences in the degree of ventilation described here, resuscitation with the pup in the thoracic position seems more appropriate. These measures should then be continued for up to 30 min p.n., depending on the condition. “

The onset point and duration of the resuscitation was not the aim of the study, therefore the conclusions are not supported by the results

The sentence line 368 was rephrased and weakened in its content:

These findings could be important in the resuscitation of neonates with neonatal respiratory depression.

33.

- line 306: as above: what is the basis of limitation of 30 minutes of resuscitation; the Authors did not examine the impact of starting point and duration of resuscitation on the pulmonary capacity and inflation

this was not the aim of the work to optimize the resuscitation methods - completely different question

Lines 328-329 “These measures should then be continued for up to 30 min p.n., depending on 328 the condition. “

I agree that it was not the aim of your study therefore why do you draw that conclusion?

This sentence was deleted.

34.

- line 312-313: this statement is not supported by the results of the study (only one puppy was delivered by CC in the study)

lines 327-328? Was rephrased to: “It can be assumed that this recommendation is particularly relevant in neonates who developed through obstetric laparotomy.”

The situation of lung development in CC pups would have to be controlled in another study, which does not get its lung fluid squeezed out per vias naturales. This would also be a very extensive further study which is not the aim of this work. - it is a crucial hint that is gratefully received but cannot be implemented in this manuscript.

I was reffering to “it would be necessary to consider whether the neonate was born per vias 335 naturalis or by obstetric laparotomy.” (lines 335-336 now)

Again, the conclusion is not supported by the results

This is not a conclusion but knowledge from literature. Thus reference [5] was included.

35.

- lines 313-318: I find it difficult to follow whether this paragraph is related to the impact of the number of foetuses on the duration of pregnancy or on the way of birth (natural or CC)

line 358: it is highly significant and results from literature – has nothing to do with way of birth only length of gestation – questionable if longer gestation results in better development of the lungs

"It has also been demonstrated that in this species, gestation length depends on the litter size [6,7]. Bitches in which the number of fetuses exceeds the normal breed average are more likely to give birth than those with a below- average number of fetuses. In boxer bitches, gestation length therefore varies considerably (55 to 71 days). Conversely, it can be assumed that the lung tissue also shows a different developmental state depending on the gestation period.” (lines 336-341 now)

Did the Authors mean that in pregancies smaller than average, the duration is prolonged, or CC is required?

The sentence was rephrased: Bitches in which the number of fetuses is above the normal breed average give birth earlier than those with a below-average number of fetuses, in which duration of pregnancy is prolonged.

36.

- lines 325 (345?): what is the relation of the presence of surfactant in the lambs' lungs to the rest of the paragraph? I find it unclear

In canine pups, surfactant protein SP-B can be detected in the lung parenchyma as early as 55 days of gestation [21].

/

37.

- lines 326-342: please add references to parts of this paragraph (and not only at the end)

Which lines are meant? Please refer to specific sentences

In 326-342 the refs [6,7] were cited which cover the complete paragraph

“At birth, some of the fetal lung fluid is forced out of the bronchial tree during thoraciccompression as the fetus passes through the maternal pelvis and vagina and is either apparently swallowed by the neonate or becomes lodged in and obstructs the upper airway“ and further on

References were included and specified to the respective sentences.

38.

- lines 343-347: according to previously mentioned doubts the conclusions are unfounded and exaggerated; basing on their study, the Authors cannot draw conclusions on the efficacy of resuscitation depending on the time of its initiation and duration

Which lines are meant? Please refer to specific sentences

I meant the conclusions. They are not supported by the methodology or results:

“The results of this study have clinical relevance in that, particularly in the context of necessary resuscitation measures, these should be performed consistently and intensively within the first 10 min p.n., preferably with the pup in the abdomen-chest position, in order to achieve the highest possible degree of ventilated pulmonary tissue. The investigations carried out here have provided the basis for evaluating the most appropriate resuscitation measures in further studies.”

39.

- References: almost half of the cited references (11 out of 23) are older than 1990 and three further are from 1990's. In my opinion newer references should be discussed to show the importance of the research to 21st century veterinary science

We have now cited Regazzi et al. and Sipriani et al. as recent papers; however, even these unfortunately only cite older literature because not much new exists

Sipriani, T. M.; Grandi, F.; da Silva, L. C.; Maiorka, P. C.; Vannucchi, C. I. Pulmonary maturation in canine foetuses from early pregnancy to parturition. Reprod Domest Anim. 2009, 44 Suppl 2, 137–140.

Regazzi, F. M.; Silva, L. C. G.; Lúcio, C. F.; Veiga, G. A. L.; Angrimani, D. S. R.; Vannucchi, C. I. Morphometric and func-tional pulmonary changes of premature neonatal puppies after antenatal corticoid therapy. Theriogenology. 2020, 153, 19–26.

Thank you for that clarification.

/

Reviewer 3 Report

The manuscript has improved considerably. Authors have made changes in the Introduction and Material and Methods sections to provide more information on the experimental design. Additionally, substantial modifications have been performed throughout.

Further comments are as follow:

Authors have justified that “However, this process is not directly related to the CT studies performed and described here.” However, I strongly disagree for that pulmonary imaging are the most important diagnostic tool to evaluate lung content, which special regard to a high sensible technique such as CT. That said, I am perfectly sure that CT images can offer accurate information on pulmonary clearance during the immediate transition period of the neonate.

Abstract

Lines 41-44: my comment is a matter of scientific writing. Your conclusion should answer this question (which is your objective): “How is the spatial and temporal development of ventilation of the lung tissue in vital canine neonates during the first 24 h p.n.?” Thus, the conclusion should be, as for example: “In conclusion, the lung tissue of canine neonates presents an aeration profile early as 10 minutes after birth and continues progressively, with a special regard to the dorsal lung areas”

Introduction:

Line 59: “Alveolar ducts and alveoli are formed postnatally [2,3,4].” The reference you show for the dog are dated in 1961. Try to use novel references that now state that alveoli can be found in recent born puppies.

Line 77-78: authors have not achieved the following investigation: “In the here presented study we investigated the lung function and the evolution of gas exchange area within the first adaptation period (0-24 h)”. Lung functioning is evaluated through pulmonary gas-exchange, which was not performed in this manuscript. Thus, my suggestion is to delete or reformulate this sentence.

Material and Methods

Clearly state in the description of the subjects the exact gestation length of the bitches. Were you dealing with any premature birth?

Clearly state the cause of death of stillbirth puppies. This information is of utmost importance to understand the CT results.

In the section “Statistical analysis”, it is fundamental to report the analysis of interaction between time of evaluation and the experimental groups. Data are presented as if a significant (p<0.05) interaction exists, i.e., the effect of body weight in each moment of evaluation, and the effect of the moment of evaluation in each group. However, if the independent variables are not under the influence of both factors simultaneously, the main effects should be considered separately. In other words, the effect of group should be analyzed merging all moments of evaluation or the effect of time should be analyzed merging all groups. Thus, authors should define which variables had a significant interaction between factors.

Include how the respiratory distress syndrome was diagnosed in Line 96.

Include detailed information on the medical procedures in Lines 96-98.

Discussion

Line 258-259: if a functional analysis of blood oxygenation was not performed, you cannot affirm that gas exchange is being fully achieved. Thus, rewrite the phrase “The aim of this study was to determine the time period, after which a sufficient gas exchange capacity can be assumed.”

Please, include your statement as a possible flaw of this study: “Because the stillborn puppies were not subjected to further pathological-anatomical examination, nothing can be said about the time of death and its cause. Only the lungs were submitted to a histopathological examination.”

Please, include reference for this phrase: “Approximately 75% of all perinatal losses in the canine species occur during or immediately after birth and thus in the 1st adaptation period (0-24 h p.n.) as well as in the initial period of the 2nd adaptation period (2nd-5th day of life).”

Please, include reference for this phrase: “As a special feature of canine neonates, their lungs can remain in an apneic inspiratory position for up to 20 seconds after the inflow of the first air stream.”

Due to the fact that you have not analyzed pulmonary gas exchange, please, consider rewriting this phrase: “It was found that already in the first 10 min after birth (expulsion) in canine neonates about 73% of the lung tissue is involved in gas exchange.”

Conclusion

My comment is a matter of scientific writing. The conclusion of the present manuscript should answer the following question (i.e. the question you were seeking to answer): “How is the spatial and temporal development of ventilation of the lung tissue in vital canine neonates during the first 24 h p.n.?” Thus, the conclusion should be the answer to the research question, as for example: “In conclusion, the lung tissue of canine neonates presents an aeration profile early as 10 minutes after birth and continues progressively, with a special regard to the dorsal lung areas”.

In the conclusion, you can restate the purpose of the experiment, identify the main findings, explain the main limitations that are relevant to the interpretation of the results and summarize what is the overall contribution of the experiment.

Author Response

The authors would like to thank reviewer #2 for his careful study and specific suggestions for improvement, which we have implemented to the best of our ability. If we disagreed we tryed to explain why.

Reviewer #3

Rebuttal 1st round

Rebuttal 2nd round

40.

The manuscript has improved considerably. Authors have made changes in the Introduction and Material and Methods sections to provide more information on the experimental design. Additionally, substantial modifications have been performed throughout.

Further comments are as follow:

44

My most important concern relates to the discussion itself. Perhaps it would better to focus your discussion on pulmonary clearance and lung content that can impair neonatal respiratory function.

The question of lung clearance is of great scientific importance. Its onset takes place during gestation (when?) and continues immediately p.n. However, this process is not directly related to the CT studies performed and described here. Indeed, the authors support the reviewer's request to investigate how lung clearance in pups obtained by caesarean section differs from that seen in pups born per vias naturales and having taken the passage through the apertura pelvis; in this Ms. this question is unresolved, not least because of the extremely small number of cases (n=1 CC), but should be urgently addressed in follow-up studies.

Authors have justified that “However, this process is not directly related to the CT studies performed and described here.” However, I strongly disagree for that pulmonary imaging are the most important diagnostic tool to evaluate lung content, which special regard to a high sensible technique such as CT. That said, I am perfectly sure that CT images can offer accurate information on pulmonary clearance during the immediate transition period of the neonate.

We apologize for the misunderstanding. Indeed, we also share completely the opinion of the reviewer and therefore included this sentence in lines 357 ff.

By pulmonary imaging lung content can be evaluated, and CT images can therefore offer accurate information on pulmonary clearance during the immediate transition period of the neonate.

However again: in this Ms. this question is unresolved, not least because of the extremely small number of cases (n=1 CC), but should be urgently addressed in follow-up studies.:

47

Line 37-38: Your conclusion is not paired with your objectives. Please, reformulate the conclusion in order to answer the scientific question established at the objective. In addition, conclusion is too overstated based on what authors have really performed in their methodology.

We apologize, but we do not understand this comment - please specify and refer to a specific sentence or statement in the Conclusions. The 10 min p.n. result directly from the determined HU measured values. Accordingly, the early resuscitation of low-life pups is justified. The appropriate body position results from the differences dorsal/ventral ventilation.

Lines 41-44: my comment is a matter of scientific writing. Your conclusion should answer this question (which is your objective): “How is the spatial and temporal development of ventilation of the lung tissue in vital canine neonates during the first 24 h p.n.?” Thus, the conclusion should be, as for example: “In conclusion, the lung tissue of canine neonates presents an aeration profile early as 10 minutes after birth and continues progressively, with a special regard to the dorsal lung areas

The conclusion was rephrased to:

Conclusion: The results of this study are in particular clinically relevant, since the lung tissue of canine neonates presents an aeration profile early as 10 minutes after birth and continues progressively, with a special regard to the dorsal lung areas.

This is the basis for  resuscitation measures that should be performed preferably with the pup in the abdomen-chest posi-tion.

Lines 60-61: New data from Regazzi et al Morphometric and functional pulmonary changes of premature neonatal puppies after antenatal corticoid therapy (2020) show that lung alveolarization starts at the final stages of fetal development. Thus, alveoli can be found in recent born puppies.

This reference was  now included in the discussion:

In dogs, cranial pulmonary lobes of the fetus are more developed than caudal lobes, as well as in a centrifugal manner [20]. Lung alveolarization starts at the final stages of fetal development [21].

Introduction

Line 59: “Alveolar ducts and alveoli are formed postnatally [2,3,4].” The reference you show for the dog are dated in 1961. Try to use novel references that now state that alveoli can be found in recent born puppies.

According to your previous recommendation we already included newer references in the discussion section.

53

Line 78: how exactly did authors perform the evaluation of gas exchange capacity? By reading the methodology, the primary postnatal function of the respiratory system to provide sufficient oxygenation and to eliminate CO2 was not analyzed in the present manuscript. Please, clarify

As part of the clinical examination of the newborns, the pH, BE and HU were correlated for time points 0,6,12,24 h - unfortunately, however, these data are not available for all 3 litters as well as for all pups. Therefore, the authors have decided not to reproduce them in this manuscript because of their incompleteness.

Line 77-78: authors have not achieved the following investigation: “In the here presented study we investigated the lung function and the evolution of gas exchange area within the first adaptation period (0-24 h)”. Lung functioning is evaluated through pulmonary gas-exchange, which was not performed in this manuscript. Thus, my suggestion is to delete or reformulate this sentence.

This sentence was reformulated:

In the here presented study we investigated the lung aeration  within the first adaptation period (0-24 h)

55

How exactly the gestation period was calculated? Based on ovulation or LH peak?

Gestation period was calculated by using vaginal cytology and P4 test before mating, from day 1 of mating the period was calculated – in Germany in principle LH peak determination is unusual and was not applied due to its inherent uncertainties.

Clearly state in the description of the subjects the exact gestation length of the bitches. Were you dealing with any premature birth?

See line 101:

the gestation period was 61.3 ±2.05 days

All three births were free of complications (birth duration Ø 473.3 min ±143.8 min; in-ter-pup interval 43.03 ± 18,4 min) and were under continuous veterinary control. That means there was none premature birth.

57

What was the cause of death of stillbirth puppies? Was a necropsy performed?

Causes of death were not determined by autopsy in the stillborn pups. Exitus letalis intra partum, klinisch frischtot

See Question #6

Clearly state the cause of death of stillbirth puppies. This information is of utmost importance to understand the CT results.

The following sentence were now included in lines 117f:

Because no additional microbiological studies were performed on the pathogenesis of stillbirth, this limits interpretation to some extent.

And in lines 302ff:

For comparison, stillborn pups were included in the first measurement period. The cause of stillbirth was not investigated and limits therefore interpretation as reference. It could be consequence of birth length, overlong inter-pup interval, but also a sequela of an infection of the dam.

Reference [14] was inserted here too.

58

The statistical analysis is too shallow. Please, include the probability values used to consider significance (P).

we do not agree that the statistical analysis is too shallow. Line 180: “As significance level p<0.05 was used.” was added to clarify

In the section “Statistical analysis”, it is fundamental to report the analysis of interaction between time of evaluation and the experimental groups. Data are presented as if a significant (p<0.05) interaction exists, i.e., the effect of body weight in each moment of evaluation, and the effect of the moment of evaluation in each group. However, if the independent variables are not under the influence of both factors simultaneously, the main effects should be considered separately. In other words, the effect of group should be analyzed merging all moments of evaluation or the effect of time should be analyzed merging all groups. Thus, authors should define which variables had a significant interaction between factors.

The authors do not agree and are convinced that the choice of statistical test satisfies the requirements to evaluate the collected data of this study. A multivariate analysis was and is not intended.

59

Line 98: how the respiratory distress syndrome was diagnosed?

Typical of NRDS is dyspnea associated with cyanotic mucous membranes and a still present heart or pulse. Apart from the typical signs of NRDS, the modified Apgar score can be used to classify the severity of NRDS in canine and feline puppies more concisely in a very short time, providing more specific indications for the emergency measures to be taken. Additional laboratory tests were performed to determine blood pH (> 7.2; 7.1-7.0; < 7.0) and current base excess (BE) (0 to +5 mmol/l; ± 0 to -10; < -10 mmol/l).

Include how the respiratory distress syndrome was diagnosed in Line 96.

This question was already answered in the first round

60

Line 99: please detail the medical procedures executed

APGAR score, BE

Include detailed information on the medical procedures in Lines 96-98

A detailed description was added as requested in lines 106ff: According to the modified Apgar score five parameter were examined to investigate the vitality of the newborn pups. Heart rate, respiratory rate, color of mucus membranes, reflex irritability and motility have been checked. A detailed clinical examination of each pup from cranial to caudal took place. The signalment (identification, color, gender, weight) was recorded and documented. The upper airways were examined for signs of obstruction. The nostrils were inspected for foam blistering. This was followed by the inspection of the eyes and ears to ensure that the systems were completely developed. The oral cavity was examined for the presence of malformations (palatochisis). The color of the mucosa, the moisture and the capillary refill time gave conclusions about the peripheral blood flow. This was followed by checking the sucking and swallowing reflex. The lungs were auscultated in all four quadrants using a Littmann stethoscope to record breath sounds. To check the cardiovascular system, the apex beat was observed, the heart was auscultated and a possible positive vein pulse of the vena jugularis was recorded. The abdomen was examined by inspection and palpation as well as recording abdominal wall tension. The umbilicus was investigated for signs of dryness and inflammation and it was palpated to rule out umbilical hernia. Finally, the correct attachment of the sexual organs, the tail and the anus was verified and the presence of an inguinal hernia was excluded.

66

Line 237-238: if a functional analysis of blood oxygenation was not performed, you cannot affirm that gas exchange is being fully achieved.

See 53. As part of the clinical examination of the newborns, the pH, BE and HU were correlated for time points 0,6,12,24 h - unfortunately, however, these data are not available for all 3 litters as well as for all pups. Therefore, the authors have decided not to reproduce them in this manuscript because of their incompleteness.

Line 258-259: if a functional analysis of blood oxygenation was not performed, you cannot affirm that gas exchange is being fully achieved. Thus, rewrite the phrase “The aim of this study was to determine the time period, after which a sufficient gas exchange capacity can be assumed.”

Was rephrased:

The aim of this study was to determine the time period, after which a sufficient aeration of the lung tissue can be assumed.

67

Line 240-241: the use of stillborn as control animals is a matter of doubt. Is it possible to affirm that fetal death occurred during whelping? Can you exclude premature fetal death? Thus, the atelectasic aspect of the lung may be only a consequence of immature lung development in a canalicular phase (please see Sipriani et al., Pulmonary maturation in canine foetuses from early pregnancy to parturition 2009)

Because the stillborn puppies were not subjected to further pathological-anatomical examination, nothing can be said about the time of death and its cause. Only the lungs were submitted to a histopathological examination

Please, include your statement as a possible flaw of this study: “Because the stillborn puppies were not subjected to further pathological-anatomical examination, nothing can be said about the time of death and its cause. Only the lungs were submitted to a histopathological examination.”

According to reviewer #1 we included the following sentence in line 306 ff:

The cause of stillbirth was not investigated in these cases and limits therefore interpretation as reference. It could be consequence of birth length, overlong inter-pup interval, but also a sequela of an infection of the dam [14].

68

Lines 255-257: this statement needs reference

Lines 280-282? Which lines are meant? The original line numbers probably got due to formatting issues in disarray. Please refer to specific sentences.

Please, include reference for this phrase: “Approximately 75% of all perinatal losses in the canine species occur during or immediately after birth and thus in the 1st adaptation period (0-24 h p.n.) as well as in the initial period of the 2nd adaptation period (2nd-5th day of life).”

This also refers to reference [14].

Please, include reference for this phrase: “As a special feature of canine neonates, their lungs can remain in an apneic inspiratory position for up to 20 seconds after the inflow of the first air stream.”

See 27. References [2,5] were included

73

Line 298: perhaps it would better to say "inflate or insufflate" instead of “gas exchange”

Actually we disagree since “gas exchange” is the usually used terminology in this context

Due to the fact that you have not analyzed pulmonary gas exchange, please, consider rewriting this phrase: “It was found that already in the first 10 min after birth (expulsion) in canine neonates about 73% of the lung tissue is involved in gas exchange.”

Rephrased to:

It was found that already in the first 10 min after birth (expulsion) in canine neonates about 73% of the lung tissue is aerated.

74

The conclusion should be mirrored by the objectives. Although very useful, future application of the data should be stated, but it cannot be considered the conclusion of the research. Please, rewrite.

The results of this study have clinical relevance in that, particularly in the context of nec-essary resuscitation measures, these should be performed consistently and intensively within the first 10 min p.n., preferably with the pup in the abdomen-chest position, in order to achieve the highest possible degree of ventilated pulmonary tissue.

My comment is a matter of scientific writing. The conclusion of the present manuscript should answer the following question (i.e. the question you were seeking to answer): “How is the spatial and temporal development of ventilation of the lung tissue in vital canine neonates during the first 24 h p.n.?” Thus, the conclusion should be the answer to the research question, as for example: “In conclusion, the lung tissue of canine neonates presents an aeration profile early as 10 minutes after birth and continues progressively, with a special regard to the dorsal lung areas”.

We modified the conclusion which reads now as follows:

In conclusion, the lung tissue of canine neonates presents an aeration profile early as 10 minutes after birth and continues progressively, with a special regard to the dorsal lung areas. These results might have clinical relevance in that, particularly in the context of necessary resuscitation measures, these should be performed preferably with the pup in the abdomen-chest position, in order to achieve the highest possible degree of ventilated pulmonary tissue. The investigations carried out here have provided the basis for evalu-ating the most appropriate resuscitation measures in further studies.

Round 3

Reviewer 1 Report

I would like to thank Authors for the changes made in the manuscript and their response to my concerns.

Reviewer 3 Report

The second revised version of the manuscript did include the requested changes and I, thus, believe it can be now accepted for publication.